# 🤔 Emoji2Idiom: Benchmarking Cryptic Symbol Understanding of Multimodal Large Language Models

## Abstract

Vision and Language are two major modalities in Artificial Intelligence research. Bridging the gap between these modalities has long been a key focus in the multimodal community. Inspired by human cognition, we believe that if a model can see an image and directly associate it with its linguistic meaning, the model possesses high-level intelligence that spans vision and language. In our work, we focus on emojis in images, a widely-used "cryptic symbol", with a data form of both visual and linguistic features, i.e. emojis have the specific textual semantics while human understand the meaning from their visual information. Specifically, we first propose the novel task of translating emojis in images to corresponding idioms, thereby challenging Multimodal Large Language Models (MLLMs) to (1) understand the semantic correlation between language and emojis, and (2) reason the intricate linguistic meaning from the emojis in images. To facilitate the advancement of this task, we construct a high-quality benchmark (`Emoji2Idiom`) following the process of automatic model generation and human manual filtering. Based on our constructed `Emoji2Idiom`, we employ multiple advanced MLLMs to conduct extensive experiments and detailed analyses, demonstrating that existing MLLMs do not yet have enough capability to understand and reason the linguistic information from visual data. We believe our proposed benchmark and interesting discoveries will encourage the community to attach importance to the intelligence of MLLMs directly associating language from vision, to give MLLMs more comprehensive vision-language understanding ability [1].

## 1 Introduction

Multimodal Large Language Models (MLLMs) have made remarkable progress and achievements in recent years Yin et al. (2023); Wu et al. (2023); Cui et al. (2024), especially their visual-language understanding capabilities, which have laid a solid foundation for the widespread development of multimodal applications Chen et al. (2024); Zhang et al. (2024a). For how to improve the visual-language understanding capabilities of MLLMs, **a core challenge is how to bridge the gap between vision and language** Koh et al. (2024); Peng et al. (2023); Wang et al. (2024).

Naturally, we want to know to what extent MLLMs should understand vision and language before we can claim that "the gap between vision and language in MLLMs has been filled"? We believe if MLLMs can behave like humans, their intelligence must have reached an extremely ideal level. We notice that when a person sees an image, he or she can often directly associate it with the linguistic meaning behind the image. For example, when humans see special symbols, they can directly know the words represented by these symbols. Since previous VQA-based benchmarks treat the vision and language separately, we try to transfer human analogy to MLLMs, when an image is fed to MLLMs, **if MLLMs can directly**

---

[1]All our data are available in anonymous Github link https://anonymous.4open.science/r/Emoji2Idiom-0CCA.

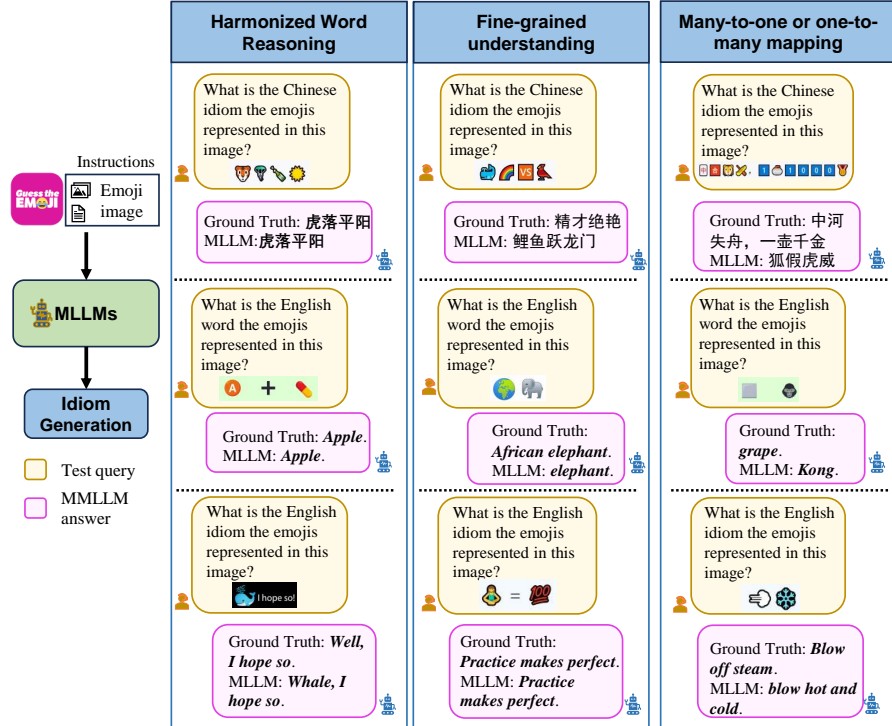

Figure 1: This figure illustrates the capabilities that our `Emoji2Idiom` concerns. We provide the emojis in images, ground truth, and the MLLM-generated results.

**associate the linguistic meaning behind the image, then we can claim that it has relatively advanced visual-language understanding intelligence.** Images inputted to MLLMs here have clear and discrete semantic meanings, manifested as concrete concepts, specific symbols or logos, etc. The semantic information of these images can be accurately interpreted as linguistic concepts or tokens.

Inspired by the above thinking, our work aims to explore the ability of MLLMs to directly understand the linguistic meaning behind images. It is exciting that many **graphic symbol codes in cryptography** exist as special indicators in an image so that when people see the image, they can decode the textual meaning of the image. We have noticed that **emoji are increasingly becoming a kind of "cryptic symbol"** widely used by people from all over the world and from all cultural backgrounds Mostafavi & Porter (2021). People not only use emoji to enrich their expressions and show their moods, but also directly use emoji to replace the corresponding text Fischer & Herbert (2021). Emojis have specific textual semantics, while human understand the meaning from their visual representations instead of directly treating emojis as characters. Thus, emojis in images are strongly coupled with their corresponding linguistic meanings, contributing to becoming the basis of our benchmarks and serving as a bridge between visual and linguistic understanding. The understanding of emoji not only requires MLLMs to comprehend the image information of individual emoji but also to combine its textual indications with the related contexts so that the model can further explore the deeper meanings of emoji. Therefore, understanding emojis in images is a challenging vision-language task.

To promote the research on cryptic symbol emoji understanding of MLLMs, we propose a novel task that requires MLLMs to receive input image information of emoji sequences and generate their corresponding text information, shown in Figure 1. We introduce the visual modality of emoji and require the MLLMs not only to identify the linguistic meaning of individual emoji but also to understand the special utterances of the emoji and its associated context, contributing to generating text with special semantic and format, e.g., a word or an idiom. Specifically, our task aims to challenge the following capabilities of MLLMs:

*(1) Harmonized Word Reasoning:* Translating emojis into texts usually harmonizes with a sound-like word. Therefore, MLLMs need to have rich language knowledge, to reason about the harmonic words.

*(2) Abstract visual understanding of image:* Emoji symbols often have strong indicative meanings, which requires MLLMs to deeply understand the abstract visual characteristics of emoji, rather than just the visual shape.

*(3) Many-to-one or one-to-many mapping generation problem:* According to our observation, it is common for multiple emojis to correspond to one word, or one emoji to correspond to multiple words. This requires the MLLMs to make correct predictions based on the origin emoji understanding and to realize the complex reasoning via context.

Furthermore, we construct the `Emoji2Idiom` benchmark to support the task of translating cryptic symbol emojis in images to corresponding texts. To enrich the diversity of the benchmark, we set up emoji-to-Chinese idiom, emoji-to-English word, and emoji-to-English idiom tasks, taking into account the language and semantic diversity. After automatic filtering and manual filtering by human experts from raw data, we obtain a high-quality dataset. From the above challenges of understanding and the design of diverse benchmark tasks, we hope that MLLM can not only realize the complex understanding of real-world text substitution expressions using emoji, but also generalize to other cryptographic symbols understanding. We hope to realize a generalized unified visual-verbal understanding benchmark instead of the traditional VQA-based benchmarks, which treat visual and verbal information separately. Based on our constructed `Emoji2Idiom`, we employ multiple advanced MLLMs to conduct extensive experiments and detailed analyses, demonstrating that existing MLLMs do not yet have enough capability to understand and reason the linguistic information from visual data. Our contributions are summarized as follows:

1. We first propose the task of translating a sequence of emojis in images to corresponding texts, aiming to guide MLLMs to perform high-level vision-language understanding like humans.

2. We build the high-quality `Emoji2Idiom` benchmark, which is a new data resource that can facilitate MLLMs to better understand cryptic symbol in images.

3. We conduct experiment of advanced MLLMs on `Emoji2Idiom` and provide some detailed analysis, interesting discoveries , and valuable insights for the community to further improve the visual-language understanding capabilities of MLLMs.

## 2 RELATED WORK

**Language Model Based Cryptic Understanding**  Emoji can be represented by UTF-8 Abel (2019), and many treat emoji as text and encode them as vectors Eisner et al. (2016). Leveraging the emoji Unicode library, numerous studies have explored emoji-text translation, including translation text into emoji Monti et al. (2016); Leonardi (2022); Klein et al. (2024), and bidirectional translationDanesi (2022). Beyond this, emoji-based sentiment analysis has become a significant area of emoji research Gibson et al. (2018); Chen et al. (2019; 2018); Liu et al. (2021). However, to the best of our knowledge, our `Emoji2Idiom` is the first to apply the visual representation and textual semantics of emojis.

**MLLMs Benchmark**  Earlier unified MLLM benchmarks collect a substantial number of images and generate corresponding QA pairs to evaluate MLLMs Fu et al. (2023a), with a focus on uniformity and objectivity, as seen in SEEDBENCH Li et al. (2024b) and SEEDBENCH-2 Li et al. (2023a). Recent benchmarks have started to assess different capabilities from different dimensions, including visual comprehension Fu et al. (2023b); Li et al. (2024a); Tong et al. (2024); Cai et al. (2023), reasoning ability Zhang et al. (2024c); Roberts et al. (2023), in-context learning capability Shukor et al. (2023); Liu et al. (2023), hallucination challenge Cui et al. (2023); Liu et al. (2023), and multiple domains (math, physics, music, medical, etc.)Lu et al. (2024b); Li et al. (2023c); Yue et al. (2024); Zhang et al. (2024b). However, most benchmarks are based on the VQA annotations and natural scenario image, rather than directly associating an abstract image with its linguistics.

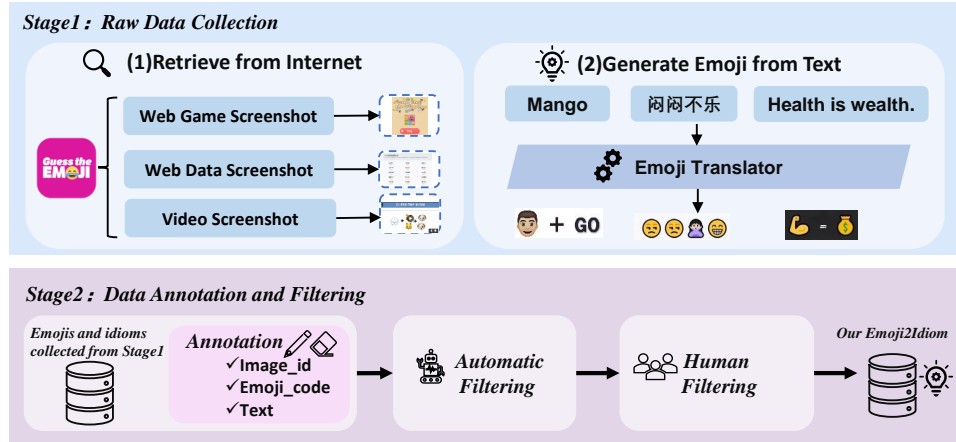

Figure 2: This figure illustrates the data collection pipeline, which is divided into two stages, raw data collection, and data annotation and filtering.

# 3 THE EMOJI2IDIOM BENCHMARK

## 3.1 TASK DEFINITION

Given an image of a sequence of emoji $I_i^{\text{emoji}} = \{\text{emoji}_1, \text{emoji}_2, \cdots, \text{emoji}_n\}$, Emoji2Idiom task aims to translate emojis in images to corresponding idiom text by model $F$:

$$\text{Text} = F(I_i^{\text{emoji}}). \tag{1}$$

These emoji sequences correspond to texts with specific formats and semantics, representing a Chinese idiom word, an English word, or an English idiom sentence. It requires not only understanding the direct corresponding text of a single emoji, but also inferring complex linguistic meaning based on the surrounding emoji context, containing some harmonic characters, multiple emoji mapping to a single character, or one emoji mapping to multiple words. These more complex emoji understanding problems are prevalent in our dataset. We further discuss the specific properties and challenges in Section 3.3 and Appendix C.

## 3.2 BENCHMARK CONSTRUCTION

**Raw Data Collection** As shown in the Figure 2, we collect raw data through two automatic generation methods: *Retrieve from the Internet.* There are a large number of databases for guessing the corresponding words and idioms based on emojis on the internet, which can be obtained freely without commercial usage. We retrieve the relevant web pages of the game, web databases, and the video to get the original emoji images and the corresponding text answers. *Generate Emoji Based on Text.* The quality of internet retrieval is not high, due to 1) recurring emoji-text pairs, 2) a relatively higher number of four-word idioms compared with a few multi-word idioms, and 3) a relatively low number of English idioms. We select the texts of common English words, English idioms, and Chinese idioms, and generate the corresponding emoji sequences by the text-to-emoji translation.

**Data Annotation and Filtering** *Automatic filtering.* The machine is utilized to automatically perform data cleaning, including deletion of duplicate values and ethical checking.

- Deletion of duplicate values and missing values, etc. Notably, the machine automatically removes duplicate combinations of the same emoji inputs but retains combinations of the same text result with different emoji inputs.
- Perform ethical checking. Some of the emoji may contain expressions of violence, pornography, or other safety violations, and we utilize GPT-4o to check all the images and remove those that are not ethically safe.

*Human Filtering.* In this phase, we engage human experts to refine the semantics of emoji-text pairs, focusing on the following aspects:

- **Removal of Non-standard Idioms:** Phrases like "蓝天白云 (blue sky and white clouds)" lack historical context and cultural significance, leading to their exclusion.

- **Elimination of Low Consistency Pairs:** Annotators assess the alignment between emoji and text. Pairs with weak correlations are discarded to make the translation process overly difficult and reducing the image's indicative meaning.

- **Exclusion of Unclear Images:** Images that are unclear to recognize, such as those from low-resolution video screenshots, are scored on their clarity. Images that score poorly are removed to ensure legibility.

- **Mitigation of Repetitive Mappings:** Frequent mappings, such as "🎀" to "结 (jie)"-"节 (jie)", can introduce data bias. To address this, we employ diverse emoji databases, and manually adjust or remove repetitive mappings beyond ten times.

- **Filtering of Unethical Content:** We rigorously filter for emoji-text pairs linked to violence, discrimination, or other inappropriate themes. A wide range of emoji including multiple skin tones and gender categories is utilized to promote expression.

To eliminate subjectivity in manual filtering, we provide annotators with detailed guidelines as shown in Appendix A. And additional information about our human filtering can be found in Appendix A.

Table 1: The statistics and image-text pair examples of our `Emoji2Idiom` in four tasks.

| Task | Image-Text Pairs | Emoji Examples | Text Examples |
|------|------------------|----------------|---------------|
| Chinese idioms (Four Characters) | 1,876 | 😟 😟 ❌ 😃 | 闷闷不乐 |
| Chinese idioms (Multi-characters) | 334 | ❌ ❓ 3 7 2 1 0 1 | 不问三七二十一 |
| English Word | 842 | ⭐ 🐟 | starfish |
| English Idiom | 783 | 💪 = 💰 | Health is wealth. |

## 3.3 Data Statistic Analysis and Other Features

We give the statistics of our proposed `Emoji2Idiom` in Table 1. In the Chinese idioms task, we collect 1,876 and 334 emoji-text pairs of four-character idioms and multi-character idioms, respectively. Among them, since the combinations of four-character idioms are naturally much larger than those of multi-character idioms in dictionaries, such a difference in the distribution is similarly reflected in our dataset. For English words, we set the tasks of emoji to word and idiom, with 842 and 783 sets of image-text pairs, respectively. There are some additional details about `Emoji2Idiom` in the Appendix B. In our `Emoji2Idiom`, we observe several interesting linguistic phenomena, and present some examples, with additional details provided in Appendix C. The linguistic phenomena raise great challenges of `Emoji2Idiom`, and also encourage the exploration of vision-language capabilities of MLLM.

**Word Split**  In the English word, it is common for multiple emoji to represent one word. For instance, the word "Panda" can be split into "Pan-" and "-da," where "Pan-" corresponds to 🔍. Beyond understanding the meaning of individual emojis, the MLLM must also remove unnecessary letters and combine the parts to infer a completely new word.

**Harmonic Characters**  Since it is sometimes difficult to find directly related emoji to represent, harmonic characters with similar pronunciations are often chosen to replace them. For example, "To be loaded", "To" harmonizes with "Two" 2, and "be" harmonizes with "bee" 🐝. In the Chinese idiom "难舍难离", "舍" harmonizes with "蛇 (snake)" of emoji 🐍, "离" harmonizes with "梨 (pear)" of emoji 🍐. The understanding of these harmonics usually requires the model to synthesize the relevant context of the emoji, to reason out the correct expression of the harmonized words.

Table 2: Evaluation results on Chinese idiom task. The `Word`, `Chr-2` and `Chr-1` denote the accuracy of guessing the whole word, two or more words, and one or more words correctly.

| | Idiom with Four words | | | | | | Idiom with Multi-words | | | | | |
| | Word-level | | | Character-level | | | Word-level | | | Character-level | | |
| Model | Word | Chr-2 | Chr-1 | Pre. | Rec. | F-1 | Word | Chr-2 | Chr-1 | Pre. | Rec. | F-1 |
|---|---|---|---|---|---|---|---|---|---|---|---|---|
| Deepseek-VL | 0.4 | 2.3 | 25.6 | 6.4 | 6.4 | 6.4 | 1.1 | 4.9 | 29.3 | 8.7 | 10 | 9.3 |
| Qwen-VL | 0.5 | 4.7 | 30.2 | 8.7 | 8.7 | 8.7 | 2.4 | 9.8 | **31.7** | **9.1** | 16.9 | **11.8** |
| LLaVa-1.5 | 0.6 | 3.8 | 32.2 | 10.5 | 10.5 | 10.5 | 2.8 | 7.9 | 29.9 | 9.0 | 17.3 | **11.8** |
| CogAgent | 0.6 | 4.4 | **34.7** | **11.6** | **11.6** | **11.6** | 3.6 | 8.2 | 30.4 | 8.7 | 14.5 | 10.9 |
| InternVL-2 | 0.8 | 6.3 | 37.8 | 9.1 | 9.1 | 9.1 | 3.4 | 8.3 | 29.4 | 8.9 | 15.6 | 11.3 |
| Claude-3.5 | 1.3 | 6.7 | 23.3 | 8.0 | 8.0 | 8.0 | 1.4 | 2.9 | 7.1 | 6.0 | 9.7 | 7.4 |
| GPT-4V | 0.7 | 1.3 | 22.1 | 5.8 | 5.8 | 5.8 | 1.1 | 6.8 | 28.4 | 3.7 | 9.1 | 5.3 |
| GPT-4o | **3.3** | **8.7** | 27.5 | 10.7 | 10.7 | 10.7 | **9.1** | **13.6** | 27.3 | 7.5 | **18.1** | 10.6 |

**Abstract visual Emoji Understanding.** In addition to referring to the direct meanings of the emoji, it is often necessary to deeply infer the semantics of the emoji. For example, in 🗑️❤️🌱💪"同心叶力 (pull together with the same goal) ", 💪 is an arm, but it does not mean "arm" in idioms. Instead, it is a very strong arm, which corresponds to "力 (power)".

**Cross-cultural Issue Discussion** Emoji, as a simple and universally recognized symbol, is widely used across many countries, especially on global social platforms. While cultural nuances are inevitable, emojis generally facilitate cross-cultural understanding. Our dataset tries to minimize ambiguities and emotional shifts caused by complex linguistic contexts, focusing on a sequence instead of a single emoji. In addition, Our dataset is constructed with careful consideration of emoji diversity, covering categories such as smiley faces and emotions, humans and bodies, animals and nature, food and drinks, travel and places, activities, objects, symbols, and flags.

## 3.4 EVALUATION METRICS

Our dataset computes the precision, recall, F-1, and BLEU value of the results with the ground truth results on the sentence level, word level, and character level to evaluate the MLLM's ability to understand emoji images. We further propose the `Chr-2` and `Chr-1` to measure in fine-grained evaluation, which denotes the accuracy of guessing two or more words, and one or more words correctly. The details about the evaluation metrics are provided in the Appendix D.

## 4 EXPERIMENT RESULTS

### 4.1 BASELINES

We select commercial Claude-3.5-sonnet-20241022, gpt-4-vision-preview and GPT-4o-20240513 to evaluate the emoji2idiom benchmark. For a richer evaluation, we select a series of open-source MLLMs for testing. These include: 1) Qwen-VL-7B Bai et al. (2023), DeepSeek-VL-7B Lu et al. (2024a), which have good **Chinese language** support; 2) LLaVa-1.5-7B Li et al. (2023b), CogAgent-18B Hong et al. (2023), InternVL-2-8B which have good **visual comprehension** capabilities. We provide a detailed description of the baselines, their implementation details, and the prompt template in Appendix E.

### 4.2 MLLM EVALUATION RESULTS

**Emoji to Chinese Idiom** We evaluate four-character and multi-character idioms shown in Table 2. We observe that all the MLLMs perform poorly on these two tasks. The latest model, GPT-4o, achieves accuracy scores of 3.3 and 5.0 at the word level for both tasks. The accuracy at the Chr-1 is significantly higher than at the word level, indicating that MLLMs are equipped with the basic translations of text corresponding to individual emojis, but have limited capability to further infer the corresponding linguistic meanings based on the

Table 3: Evaluation on English word and idiom task. B-1 and B-2 denote the BLEU-1 and BLEU-2 respectively.

| | English Word | | | | | | English Idiom | | | | | | | |
| | Word-level | | | Character-level | | | Sentence-level | | | Word-level | | | | |
| Model | Pre. | Rec. | F-1 | Pre. | Rec. | F-1 | Pre. | Rec. | F-1 | Pre. | Rec. | F-1 | B-1 | B-2 |
|---|---|---|---|---|---|---|---|---|---|---|---|---|---|---|
| Deepseek-VL | 23.2 | 26.3 | 24.7 | 46.2 | 47.5 | 46.8 | 11.9 | 11.9 | 11.9 | 15.1 | 14.6 | 14.8 | 15.1 | 11 |
| Qwen-VL | 28.6 | 29.1 | 28.8 | 51.2 | 50.4 | 50.8 | 12.1 | 12.1 | 12.1 | 17.1 | 12.5 | 14.4 | 17.1 | 11.3 |
| LLaVa-1.5 | 30.1 | 30.1 | 30.1 | 54.6 | 55.7 | 55.1 | 14.4 | 14.4 | 14.4 | 19.7 | 21.3 | 20.5 | 19.7 | 16.4 |
| CogAgent | 29.8 | 29.8 | 29.8 | 52.8 | 51.9 | 52.3 | 13.2 | 13.2 | 13.2 | 18.3 | 19.5 | 18.9 | 18.3 | 15.2 |
| InternVL-2 | 31.1 | 31.1 | 31.1 | 56.6 | 57.2 | 56.9 | 15.3 | 15.3 | 15.3 | 19.3 | 22.1 | 20.6 | 18.4 | 16.1 |
| Claude-3.5 | 42.3 | 42.3 | 42.3 | 63.9 | 73.8 | 68.5 | 29.8 | 29.8 | 29.8 | 48.0 | 42.7 | 45.2 | 42.3 | 39.7 |
| GPT-4V | 38.5 | 38.5 | 38.5 | 60.3 | 69.2 | 64.4 | 26.4 | 26.4 | 26.4 | 41.1 | 43.1 | 42.1 | 39.4 | 37.5 |
| GPT-4o | **55.8** | **55.8** | **55.8** | **68.5** | **77.5** | **72.7** | **35.2** | **35.2** | **35.2** | **46.8** | **47.3** | **47.0** | **45.0** | **41.6** |

Table 4: Evaluation of the semantic similarity scores of Chinese task, where 1 is categorized as dissimilar and 5 is categorized as perfect similarity.

| | Chinese Idiom with Four words | | | | | | Chinese Idiom with Multi-words | | | | | |
| | Average | Distribution(%) | | | | | Average | Distribution(%) | | | | |
| Model | Semantics | 1 | 2 | 3 | 4 | 5 | Semantics | 1 | 2 | 3 | 4 | 5 |
|---|---|---|---|---|---|---|---|---|---|---|---|---|
| InternVL-2 | 1.41 | 66.9 | 26.3 | 6.1 | 0 | 0.7 | 1.47 | 61.4 | 32.9 | 4.5 | 0 | 1.1 |
| GPT-4o | 1.66 | 56.7 | 29.1 | 9.5 | 0.6 | 4.1 | 1.76 | 59.1 | 25.0 | 5.7 | 1.2 | 9.0 |
| | English Word | | | | | | English Idiom | | | | | |
| InternVL-2 | 2.75 | 46.2 | 11.5 | 19.2 | 17.2 | 40.4 | 2.75 | 60.3 | 19.0 | 11.1 | 4.8 | 4.9 |
| GPT-4o | 3.55 | 27.5 | 7.8 | 2.0 | 7.5 | 55.2 | 2.99 | 28.5 | 22.0 | 7.7 | 5.5 | 36.3 |

relevant emoji context, especially for the harmonization reasoning. Thus, our `Emoji2Idiom` is challenging to MLLMs due to a huge number of harmonization word mapping with emojis.

**Emoji to English Word and English Idiom**   MLLM's overall accuracy is higher compared to the two Chinese idiom tasks. In Table 3, GPT-4o achieves impressive F-1 values of 55.8 and 35.2 at the word and sentence levels, in emoji-to-English word and English idiom respectively. This is likely because the model has encountered more similar English texts during training, making it more adept at reasoning about English words. However, MLLMs always suffer from hallucination problems. When they catch a linguistic meaning of a single emoji, they quickly focus on the word or idiom related to this emoji from the inner knowledge they have, and ignore the relevant context of the emojis. Based on our `Emoji2Idiom`, the community can explore the hallucination problem and improve the inference ability.

**Evaluation of the semantic similarity of the response**   We further experiment the semantic similarity between the responses and the ground truth. We input the model output answers and ground truth into LLM and let LLM score the semantic similarity from 1 to 5. As shown in the Table 4.2, we observe that the average scores of the model on the English task are significantly higher than those on the Chinese task. In addition, when we carefully observe the distribution, we find that 1)for the Chinese task, most of the scores are concentrated in 1 and 2, which indicates that the MLLM can almost barely guess; 2)while for the English task, most of the scores are concentrated in 1 and 5, which indicates that the MLLM can either predict the answer correctly, or get irrelevant answers.

### 4.3 FURTHER EXPLORATION ON MLLM LEARNING

**Exploration with In-context Learning**   In addition to evaluating the direct inference abilities of MLLMs, we further explore their performance using in-context learning. We select the open-source Qwen-VL and the closed-source GPT-4o, evaluating each task with 3, 5, and 7 context examples, as shown in Table 5 and Table 6. MLLMs improve across various tasks with the addition of contextual examples, indicating the high quality of our `Emoji2Idiom` that the randomly chosen examples can improve the performance a lot. However, in the Chinese task, performance decreases when using too many samples (7 in-context examples). This decline indicates that MLLMs learn incorrect mappings in this complex

Table 5: Exploration on in-context learning in Chinese idiom tasks. The `Word`, `Chr-2` and `Chr-1` denote the accuracy of the whole word, two or more words, and one or more words.

| | Idiom with Four words | | | | | | Idiom with Multi-words | | | | | |
| | Word-level | | | Character-level | | | Word-level | | | Character-level | | |
| Model | Word | Chr-2 | Chr-1 | Pre. | Rec. | F-1 | Word | Chr-2 | Chr-1 | Pre. | Rec. | F-1 |
|---|---|---|---|---|---|---|---|---|---|---|---|---|
| Qwen-VL | 0.5 | 4.7 | 30.2 | 8.7 | 8.7 | 8.7 | 2.4 | 9.8 | 31.7 | 9.1 | 16.9 | 11.8 |
| +3 in-context example | 0.5 | 5.1 | 31.3 | 9.3 | 9.3 | 9.3 | 2.2 | 10.1 | 28.6 | 10.4 | 13.1 | 11.6 |
| +5 in-context example | 0.6 | 5.3 | 31.6 | 9.4 | 9.4 | 9.4 | 3.3 | 12.3 | 32.1 | 11.7 | 16.9 | 13.8 |
| +7 in-context example | 0.5 | 4.9 | 32.1 | 9.4 | 9.4 | 9.4 | 2.8 | 10.7 | 31.4 | 11.4 | 15.4 | 13.1 |
| GPT-4o | 3.3 | 8.7 | 27.5 | 10.7 | 10.7 | 10.7 | 9.1 | 13.6 | 27.3 | 7.5 | 18.1 | 10.6 |
| +3 in-context example | 2.6 | 11.3 | 33.9 | 12.6 | 12.6 | 12.6 | 9.5 | 23.8 | 36.9 | 17.0 | 21.9 | 19.1 |
| +5 in-context example | 3.5 | 12.2 | 35.7 | 13.7 | 13.7 | 13.7 | 13.1 | 27.4 | 42.9 | 20.7 | 29.1 | 24.2 |
| +7 in-context example | 3.5 | 8.7 | 31.3 | 12.0 | 12.0 | 12.0 | 10.7 | 19.0 | 34.5 | 16.2 | 23.1 | 19.0 |

Table 6: Exploration on in-context learning in English tasks.

| | English Words | | | | | | English Idiom | | | | | |
| | Word-level | | | Character-level | | | Word-level | | | Character-level | | |
| Model | Pre. | Rec. | F-1 | Pre. | Rec. | F-1 | Pre. | Rec. | F-1 | Pre. | Rec. | F-1 |
|---|---|---|---|---|---|---|---|---|---|---|---|---|
| Qwen-VL | 28.6 | 29.1 | 28.8 | 51.2 | 50.4 | 50.8 | 12.1 | 12.1 | 12.1 | 17.1 | 12.5 | 14.4 |
| +3 in-context example | 29.3 | 29.3 | 29.3 | 53.6 | 52.1 | 52.8 | 12.0 | 12.0 | 12.0 | 17.6 | 17.9 | 17.7 |
| +5 in-context example | 30.6 | 30.6 | 30.6 | 55.9 | 54.2 | 55.0 | 13.0 | 13.0 | 13.0 | 18.9 | 18.4 | 18.6 |
| +7 in-context example | 32.5 | 32.5 | 32.5 | 57.8 | 55.7 | 56.7 | 15.2 | 15.2 | 15.2 | 19.7 | 20.2 | 19.9 |
| GPT-4o | 55.8 | 55.8 | 55.8 | 68.5 | 77.5 | 72.7 | 35.2 | 35.2 | 35.2 | 46.8 | 47.3 | 47.0 |
| +3 in-context example | 57.6 | 57.6 | 57.6 | 72.3 | 75.0 | 73.6 | 36.3 | 36.3 | 36.3 | 47.6 | 47.3 | 47.4 |
| +5 in-context example | 54.5 | 54.5 | 54.5 | 77.5 | 79.0 | 78.2 | 37.4 | 37.4 | 37.4 | 48.2 | 50.0 | 49.1 |
| +7 in-context example | 60.6 | 60.6 | 60.6 | 79.4 | 73.9 | 76.5 | 38.5 | 38.5 | 38.5 | 49.5 | 50.5 | 50.0 |

task and suffer from hallucination issues. We further provide some insights of fine-tuneing and reasoning approaches on `Emoji2Idiom` in Appendix. I.

**Exploration with Chain-of-Thought** We further investigate the enhancement of CoT inference. This task prompts the MLLM to think step by step, with the detailed prompt in the Appendix E.3.2. In Figure 3, we evaluate GPT-4o and qwen-vl. Our findings indicate that the CoT design enables the MLLM to produce better answers without additional training, improving accuracy at both character and word levels while improving semantically and visually aligned responses. **This demonstrates the method's effectiveness and the high quality of our data.** Furthermore, the CoT framework mitigates hallucination issues in GPT-4o while avoiding significant semantic bias. By mimicking human reasoning processes, the CoT design offers insights into MLLM errors, guiding future research.

**Exploration on Input Length Effects** We discuss that how the length of emoji sequences might impact model performance. The length of chinese four-character idioms and English words exhibit short, with average lengths of 4.11 and 4.23, respectively, and lead to minimal impact from image size variations. However, Chinese multi-character idioms and English idioms have longer sequences (averaging 7.48 and 5.32), resulting in more elongated images with higher variance in length. The resizing methods employed by different MLLMs can distort longer images, degrading performance. To address this, further work can propose an additional preprocessing step.

## 4.4 HUMAN EVALUATION

**Human Performance on Chinese Idiom Tasks** Due to the limited performance of MLLMs on the Chinese idiom task, we invite human experts to participate and assess the task's difficulty, thereby determining the upper limit of machine performance on this benchmark. Humans are tested by the same evaluation metrics, and task complexity is rated on a scale from one (very easy) to five (very difficult), with evaluation details provided in Appendix G. The results in Figure 4 show that MLLM still has significant room for improvement, and our `Emoji2Idiom` presents significant challenges.

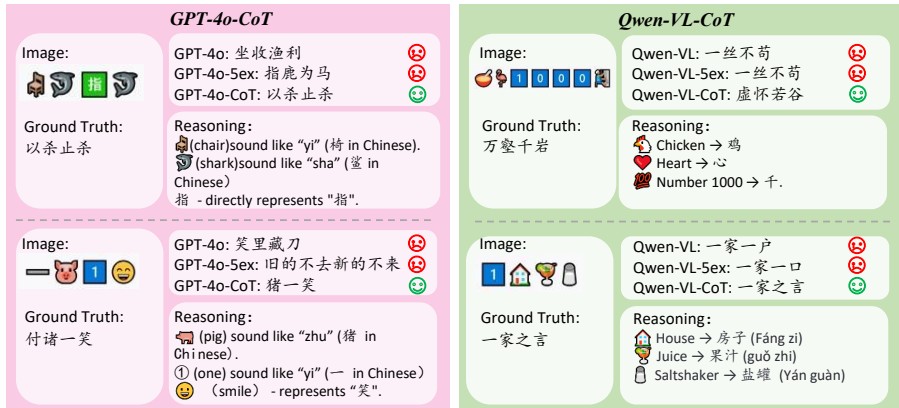

Figure 3: The results of the base MLLM, ICL approach, and CoT approach, with the reasoning process of MLLM.

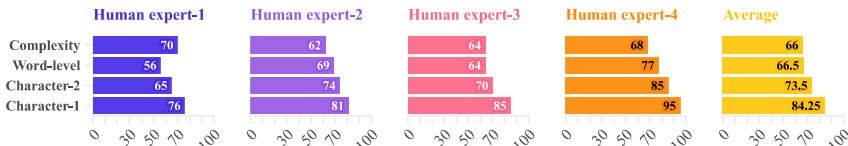

Figure 4: Human performance on Chinese idiom task. To better show the task complexity, we map the score to the 1-100 interval.

**Human Evaluation on MLLMs**  We conduct a human assessment of the answers generated by MLLMs. This evaluation considers the normality, semantic similarity to the ground truth, emotional similarity, visual similarity to the original emojis, and the fluency of the generated text, with details provided in Appendix. G. As shown in Figure 5, MLLMs perform well in generating standardized idiom expressions. However, lower scores in semantic similarity and visual similarity suggest that emoji comprehension and idiom reasoning remain challenging areas for MLLMs.

4.5 CASE STUDY

We provide a case study based on our experimental results, which contains four types of prevalent challenges, and propose some potential training methods in Appendix I.

**Harmonization Problem**  MLLMs often fail on harmonization problems. As shown in Figure 6, 🐍 - "蛇 (snake, sound like "she")" homonym to "舍 (leave, sound like "she")" but the MLLM fails to recognize. In the English idiom, 🐳 - "whale" homonym to "well". Our Emoji2Idiom includes many harmonic character phenomena. Current MLLMs are not yet capable of effectively capturing emoji context and reasoning with harmonic words, and struggle with our challenging Emoji2Idiom.

**Hallucination Problem**  In Figure 6, the model recognizes 🐎 and immediately outputs "horsing around", without considering other emojis. Another example shows GPT-4v and GPT-4o recognizing the number 3️⃣ and associating it with the idioms "朝三暮四 (change one's mind often)" and "颠三倒四 (disorderly)", both containing the number 3, without considering the surrounding emojis. That is due to the hallucination problem. MLLMs often think narrowly, focusing only on words or idioms directly related to a single emoji. Our Emoji2Idiom is concerned about this issue, and look forward to further exploration of the poor performance of MLLMs that we have discovered.

**Multi-emoji to One Character Mapping.**  Emoji2Idiom presents a huge challenge on this mapping issue, where MLLMs fail to perform a multi-to-one or one-to-multi mapping.



Figure 5: Human evaluation on GPT-4v and GPT-4o. The Std., Sem Sim., Emj Sim., Emo Sim., and Flu. denote the normality, semantic similarity, emotional similarity, visual similarity to the original emojis, and fluency.

| Problems | Chinese Idiom with four character | Chinese Idiom with Multi-characters | English Word | English Idiom |
|---|---|---|---|---|
| Couldn't identity *homophonic characters*（Red color denotes the homophonic characters） | Emoji:

Ground truth: 难舍难离
GPT-4V: 南瓜蛇果
GPT-4o: 南瓜蛇离 | Emoji:

Ground truth: 捷雷不及掩耳
GPT-4V: 闻鸡起舞
GPT-4o: 闻鸡起舞 | Emoji:

Ground truth: kiwi
GPT-4V: keywest
GPT-4o: keyword | Emoji:

Ground truth: Well, I hope so
GPT-4V: whale of a time
GPT-4o: Whale, I hope so |
| Suffer *hallucination* problem | Emoji:

Ground truth: 以肉喂虎
GPT-4V: 狐假虎威
GPT-4o: 如坐针毡 | Emoji:

Ground truth: 不问三七二十一
GPT-4V: 三心二意
GPT-4o: 颠三倒四 | Emoji:

Ground truth: Killer whale
GPT-4V: swordfish
GPT-4o: swordfish | Emoji:

Ground truth: To pony up
GPT-4V: straight from the horse's mouth
GPT-4o: horsing around |
| *Multi-emoji to one* character mapping | Emoji:

Ground truth: 万壑千岩
GPT-4V: 一心一意
GPT-4o: 差强人意 | Emoji:

Ground truth:中河失舟，一壶千金
GPT-4V: 朝三暮四
GPT-4o: 狐假虎威 | Emoji:

Ground truth: Blackberry
GPT-4V: Bare
GPT-4o: Squarebear | Emoji:

Ground truth: Blow off steam
GPT-4V: cold shoulder
GPT-4o: blow hot and cold |
| *Abstract Visual understanding* of the emoji symbol | Emoji:

Ground truth: 精才绝艳
GPT-4V: 东施效颦
GPT-4o: 鲤鱼跃龙门 | Emoji:

Ground truth:知无不言，言无不听
GPT-4V: 人山人海
GPT-4o: 见不得人 | Emoji:

Ground truth: African elephant
GPT-4V: worldwide
GPT-4o: elephant | Emoji:

Ground truth: Receive a kickback
GPT-4V: think outside the box
GPT-4o: out of the box |

Figure 6: Four typical problems the GPT-4v and GPT-4o suffer in our `Emoji2Idiom`.

For example, 1 0 0 0 are four emojis, but the model does not successfully combine them into one character "千 (one thousand)".

**Abstract Visual Image Understanding of the Emoji Symbol**  MLLMs struggle to align emoji semantics with intricate meanings when it comes to deep comprehension. For example, in "receive a kickback", the model simply captures 📦, the meaning of "box", and interprets it as "out of the box", but does not combine the package attributes of "receiving something" with the hint of money to generate the correct answer. Our `Emoji2Idiom` highly focuses on this deeper understanding, evaluating and exploring the capabilities of MLLMs.

## 5 CONCLUSION

We propose the `Emoji2Idiom` benchmark, containing emoji to Chinese idioms, English words, and English idioms. It provides a way to measure the ability of MLLM to understand complex emoji symbol sequences on images. We design a measurement framework containing harmonic characters, abstract visual understanding, and many-to-one mapping problems, to validate the ability of the MLLM to synthesize the understanding of emoji contexts with emoji-to-text coupled reasoning and generation. We evaluate advanced open-source and closed-source MLLMs with our dataset, analyze the results, and highlight future research directions with case studies.

## 6    Reproducibility Statement

In order to ensure that other researchers can better reproduce our work in us, we put a lot of effort into reproducibility. We describe in detail our data collection and data building process in Section 3 and Appendix A, and provide full experimental details in the Section 4.1 and Appendix E,G, including the parameter details of the model we used with the prompt template. All data and source code can be found on the Github link Emoji2Idiom. We promise to continue to maintain our Github repository, discuss this research with other researchers, and contribute to the entire multimodal large language model community.

## 7    Ethics Statement

We introduce a novel benchmark, `Emoji2Idiom`, incorporating a thorough description of data collection, annotation, and filtration processes. We emphasize that the dataset's creation adheres strictly to ethical guidelines, with vigilant measures against any breach or impropriety. Great care has been taken to uphold ethical standards in the dataset, employing anonymization, desensitization, and data cleaning. The text samples pose no risk to public welfare. Hence, the innovative research directions and tasks proposed are ethically robust and harmless to society.

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

# A    ADDITIONAL DETAILS OF DATA FILTERING

## A.1    AUTOMATIC FILTERING

In this phase, we mainly utilize machines and large language models to filter large-scale data, which includes the following steps in total:

1. Deletion of default values. We utilize a machine to automatically remove incomplete emoji-idiom pairs, including those with missing corresponding emoji images and those with missing corresponding idioms. It is guaranteed that each emoji image corresponds to the standard idiom answer one by one.

2. Removing Duplicate Values. We utilize the machine to automatically remove duplicate emoji-idiom pairs. Here, we only need to remove the emoji-idiom pairs corresponding to identical emoji sequences while retaining the pairs with the same idiom text but corresponding to different emoji representations, which helps to enhance the diversity of the dataset. Note that we will first filter the pairs corresponding to the same idiom text by the machine with additional labels, and make a manual decision on whether to perform the deletion in the next stage of manual filtering.

3. Image Quality Check. We utilize LLM (specifically GPT-4o is used), to perform image quality checking, which entails marking and removing: images that are too blurry and those that do not meet the ethical norms (images that contain elements of violence, abusive language, discrimination, etc.) along with their corresponding idioms.

4. Text Ethics Checking. We utilize LLM (specifically GPT-4o) to perform text ethics checking, which involves tagging and deleting idiom with elements of violence, discrimination, abuse, etc. For example, "红颜祸水" is a sexist idiom, and we will delete its corresponding emoji-idiom pair.

## A.2    HUMAN FILTERING

In this phase, we invited human experts in Chinese and English languages to perform manual data filtering, which included the following steps in total:

1. Duplicate value checking: for the automatic filtering phase, the machine flags a portion of emoji-idiom pairs where the text is the same but the corresponding images are not the same. the human expert needs to further check whether the emoji expressions here are really different. For the pairs with identical emoji images, the human expert will delete them.

2. Image quality check: Human experts further check whether the emoji images are unclear and illegible, and remove the illegible images.

3. Idiom standardization check: Human experts need to check whether the idiom text expression is standardized, including the format of the idiom, whether it has a specific linguistic meaning, and whether it is in line with common human usage, etc., to ensure that our dataset meets the real-world usability. For example, for the idiom "blue sky and white clouds", although it is a four-word idiom that conforms to the norms of human usage, it does not have a specific allusion, mythological story, traditional story background, or special semantic meaning, and does not belong to the standard idioms. For example, although "流水高山" is a four-letter word with a specific historical background, people more often use the expression "high mountains and flowing water". Therefore, "流水高山" is not an expression that conforms to human language usage and will be deleted.

4. Emoji and Idiom Relevance Check: Since in emoji to idiom expression, many times the representation of harmonic characters will be utilized, which will increase the difficulty of emoji comprehension and the difficulty of generating the final idioms. Human experts will evaluate the relevance of emoji to idioms:

    - If too many or too complex harmonic characters are used with the emoji representation, at this time the task will be too difficult for not only MLLM but

also humans to understand. At this point, the human expert will consider the relevance of this emoji sequence to the idiom to be too low and delete the emoji-idiom pair.

- It is noteworthy that we conducted an evaluation of human ability on this benchmark in Sec. 4.4 and found that humans achieved an average score of 66.5 on the word-level accuracy of Chinese idioms. This score demonstrates both that our dataset is challenging and that the task is accomplishable, and that there is still much room for improvement in the performance of the current MLLM on this task.

5. Repeated harmonic word mapping check: due to the limited expression of emoji, when using emoji to replace textual expressions, harmonic words are often used to find the corresponding emoji for expression. emoji2idiom also has a large number of harmonic words. However, we found that if the mapping of the same emoji corresponding to a certain harmonic word occurs too many times, it may cause data bias to LLM in subsequent training, i.e., when LLM sees this emoji it automatically thinks of this harmonic word that occurs multiple times. To mitigate the bias caused by this harmonic word mapping, we performed:

   - Count the repeated emoji-character harmonic word mappings, and when there are more than ten occurrences, we manually replace the expression of the emoji (find other harmonic word counterparts to replace the original repeated emoji), or just delete the redundant emoji-idiom pair.
   - In addition, we also considered this issue during the original data collection. Our retrieval and collection in different sources of the original emoji database can reduce this duplicate mapping. We also take different generation methods when manually constructing text-to-emoji data, which also helps to increase the diversity of harmonic word mappings.

6. Safety and Ethics Check: Based on the automatic detection, the human experts further conducted a safety and ethics check of the emoji images and idiom text, checking whether there are any issues such as violent gore, abusive language, sexism, racial discrimination, stereotyping, and so on, in the data.

To eliminate subjectivity in manual filtering, we provide annotators with detailed guidelines as shown in Figure 7 and 8, including scoring criteria for each item (1-5 points) covering idiomatic normality, graphic consistency, image legibility, repetition mapping, and ethical safety checks. We also provide at least three examples for each item. For ethical safety checks, we distinguish between subcategories such as violence, abusive language, gender discrimination, stereotyping, and racial discrimination. We provide examples at both the emoji and text levels to guide judgments.

## B  ADDITIONAL DETAILS OF DATA STATISTICS

In addition to the numerical statistics, we further do some statistics to better show our `Emoji2Idiom`.

**Word Frequency and Word Cloud Statistic of Chinese idiom**    To better present our dataset, we perform word frequency statistics on Chinese idioms and display the word cloud and word rectangle tree graphs, as shown in Figure 9. We first perform word frequency statistics on all characters, filter out the top 1,000 characters, and discard low-frequency words. From the filtered top 1,000 characters, we conduct lexical analysis and plot word cloud and word rectangle diagrams for adjective and adverbial morphemes, noun morphemes, and verb morphemes, respectively.

**Word Frequency and Word Cloud Statistic of English idiom**    Similarly, we perform word frequency statistics on English idioms and display word cloud maps with word rectangle tree diagrams, as shown in Figure 10. We first perform word frequency statistics on all words, filter out the top 180 words, and discard low-frequency words. From the filtered top 180 words, we create word cloud maps with word rectangle mapping.

| Guideline of Human Filtering of Data - 1 | |
|---|---|
| The purpose of this work is to screen out emoji-text pairs that do not comply with the rules. For each indicator, there will be a corresponding criterion and examples, which you will need to score the emoji-text pairs, and only the pairs that meet the requirements of each indicator can be retained. | |
| **Case** | |
| **Image:**

🎀🌰👱🦊👤👂

**Text：捷雷不及掩耳** | Normality: 5
Consistency: 4
legibility: 4
Emoji Ethical security: 4
Text Ethical security: 5 |
| **Metrics** | |

> **Normality:** Whether the text conforms to the idiom's specifications. This includes whether it has historical allusions and specific cultural backgrounds, or does not conform to human usage habits

| Options | 1. Completely non-standard  2. Mostly non-standard  3. Fairly standard
4. Mostly standard  5. Completely standard |
|---|---|
| Examples | 1.  "朝三暮四" shows completely standard to the normality.
2.  "蓝天白云" shows completely non-standard.
3.  "红红火火" shows mostly non-standard. |

> **Consistency**: The consistency of the emoji and the image is scored, and the higher the consistency of the example, the easier it is to get the final translation result

| Options | 1. Completely inconsistent  2. Mostly inconsistent  3. Fairly consistent
4. Mostly consistent  5. Completely consistent |
|---|---|
| Examples | 1.  2️⃣🖐3️⃣🔪 -"两面三刀" pair is mostly consistent.
2.  🔺🅿🔁👂 -"出尔反尔" pair is mostly inconsistent.
3.  🐵 - "I'm bored." pair is completely inconsistent. |

> **legibility:** Whether the image is very blurry and illegible is difficult for MLLM to process.

| Options | 1. Completely illegible  2. Mostly illegible  3. Fairly legible
4. Mostly legible  5. Completely legible |
|---|---|
| Examples | 1.  🍪🍙🍎🔪 shows completely legible.
2.  🧀 + 🐜 shows mostly illegible.
3.  D + 📗 + 🔑 shows fairly legible. |

> **Duplicate emoji-character mapping:** Remove or modify the duplicate emoji-character mapping of emojis.

| Examples | 🔑🐚🔳🎋  玉宇琼楼  🔺🌸1️⃣🔍  玉减香消  ◯🐚😊🐒  白玉微瑕
🚪🔔💎🔪  钟鼓馔玉  🐱🐚👤◯  金玉其表  🎖🐓🐚🐖  金风玉露
In these examples, 🌽 (corn, 玉米) is used to map the Chinese character "玉" |
|---|---|
| Method | Count a single emoji and a single character pair that occur repeatedly. When there are more than 10 times, delete the corresponding emoji-text pairs, or replace a single emoji until the number of homophonic pairs is equal to 10. |

Figure 7: The first page guidelines for human filtering.

| Guideline of Human Filtering of Data - 2 | |
|---|---|
| The purpose of this work is to screen out emoji-text pairs that do not comply with the rules. For each indicator, there will be a corresponding criterion and examples, which you will need to score the emoji-text pairs, and only the pairs that meet the requirements of each indicator can be retained. | |
| **Case** | |
| **Image:**

🎀🌂👥🐓👤👂

**Text:** 捷雷不及掩耳 | Normality: 5
Consistency:  4
legibility:  4
Emoji Ethical security:  4
Text Ethical security:   5 |
| **Evaluation Metrics** | |

| ➢ **Ethical security check:** Remove emoji-text pairs that are not ethically safe. We rigorously vet emoji-text pairs for issues such as violence, name-calling, and gender bias. | |
|---|---|
| Emoji images filtering | |
| **Possible issues** | Contains elements of violence, abusiveness, racial discrimination, gender discrimination, and stereotypes. |
| **Options** | 1.   Completely insecure   2. Mostly insecure   3. Fairly secure
4.   Mostly secure   5. Completely secure |
| **Examples** | 1.   🍑2👥B   shows completely insecure to the ethics, due to the abusiveness.
2.   31 9 👥😄   shows mostly insecure to the ethics, because it does not conform to social order and good customs.
3.   👁👥👀📇🐎   shows completely insecure to the ethics, due to violence. |
| Texts filtering | |
| **Possible issues** | Contains elements of violence, abusiveness, racial discrimination, gender discrimination, stereotypes, expressions of partiality and passion, and does not conform to social order and good customs. |
| **Options** | 1.   Completely insecure   2. Mostly insecure   3. Fairly secure
4.   Mostly secure   5. Completely secure |
| **Examples** | 1.   "头发长见识短" shows mostly insecure to the ethics, due to the gender discrimination on the women.
2.   "男主外女主内" shows completely insecure to the ethics, due to the stereotypes.
3.   "feisty woman" shows completely insecure to the ethics, due to the gender discrimination. |

Figure 8: The guidelines for human filtering.

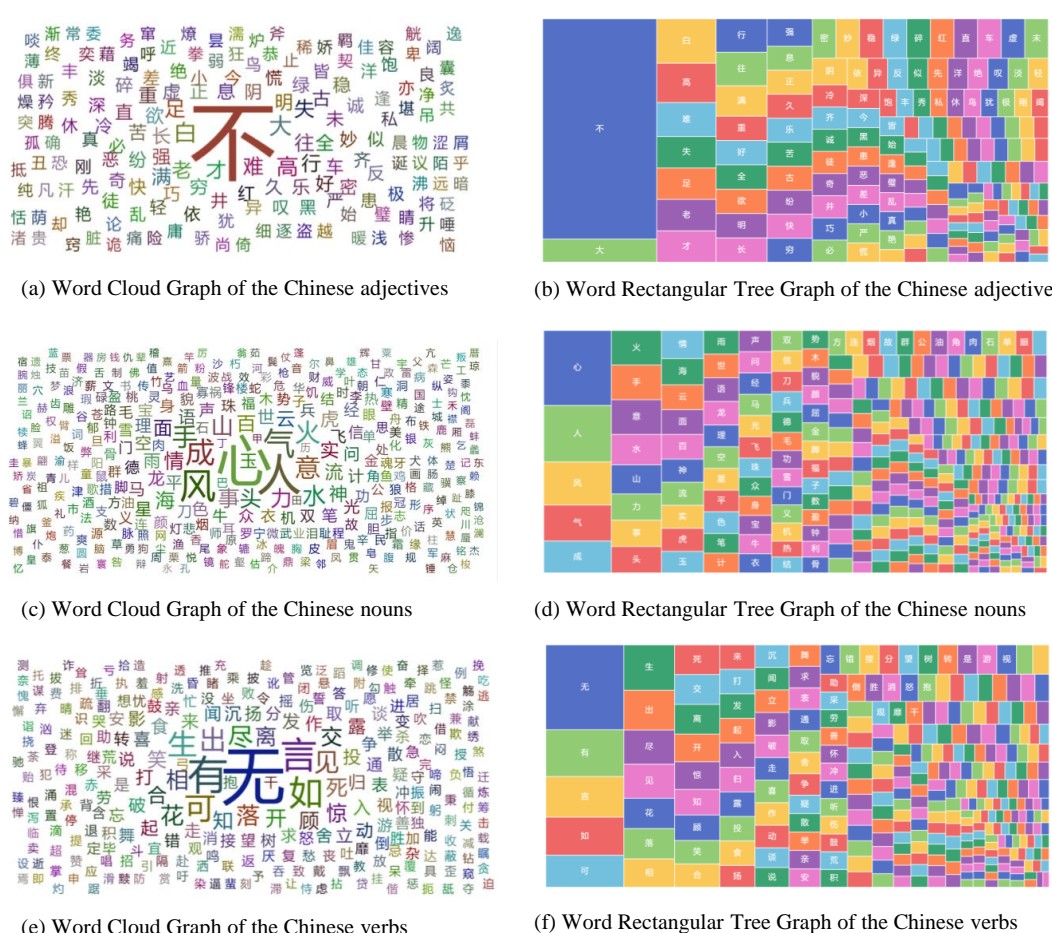

(a) Word Cloud Graph of the Chinese adjectives

(b) Word Rectangular Tree Graph of the Chinese adjectives

(c) Word Cloud Graph of the Chinese nouns

(d) Word Rectangular Tree Graph of the Chinese nouns

(e) Word Cloud Graph of the Chinese verbs

(f) Word Rectangular Tree Graph of the Chinese verbs

Figure 9: Word cloud and word rectangle diagrams of Chinese idiom, including adjective and adverbial morphemes, noun morphemes, and verb morphemes.

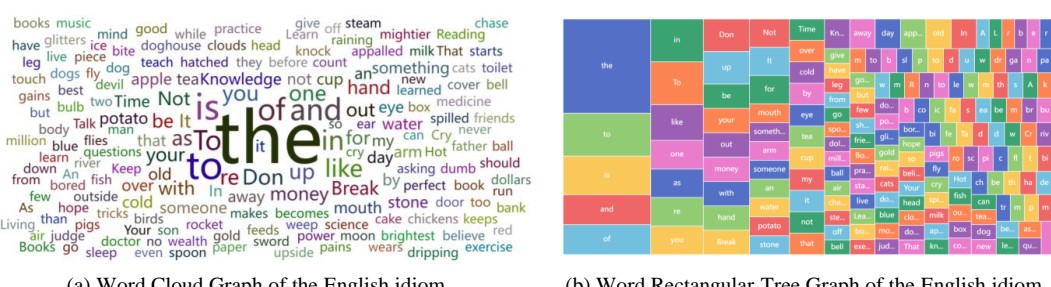

(a) Word Cloud Graph of the English idiom

(b) Word Rectangular Tree Graph of the English idiom

Figure 10: Word cloud and word rectangle diagrams of English idiom.

## C Additional Details of Data Attributes and Linguistic Phenomenon

### C.1 Chinese Idiom Task

**Harmonization Word**  Since it is sometimes difficult to find directly related emoji to represent, harmonic characters with similar pronunciations are often chosen to replace them. For example, Usually, for characters that can't be represented directly by emoji, we will first search for their harmonized characters, then find an emoji that can directly represent the harmonized character, and replace it with this emoji. For example, "捷" does not have a direct emoji, but it harmonizes with 结", which corresponds to "bow" 🎀, and so, we select 🎀 chosen to represent the character "捷". There are a large number of harmonic characters in our data. This poses a great challenge to MLLM's understanding and reasoning ability. The reasoning of harmonic words needs the help of related contexts, and in our data scenario, the model is required to analyze the context of emoji in depth instead of understanding individual emoji alone. The understanding of these harmonics usually requires the model to synthesize the relevant context of the emoji, to reason out the correct expression of the harmonized words.

### C.2 Abstract visual Emoji Understanding.

The model shows better performance in simply recognizing the shallow meanings of individual emoji, but in Abstract visual in-depth understanding, it is difficult for the model to work with the contextual emoji information to get the real corresponding relevant emoji meanings. For example, 🆚 means match, PK, duel, competition, and so on. In Chinese, "决" represents duel, and then harmonized to "绝" to get the idiom "精才绝艳". In "African elephant", the superficial meaning of the emoji is the earth, but further combined with the specific location of the earth map in the figure and the hint of an elephant, the emoji represents the African elephant. Abstract visual understanding in conjunction with its textual meaning to further reason about the correct answer. In 🎰❤️🍃💪"同心叶力 (pull together with the same goal) ", 💪 is an arm, but it does not mean "arm" in idioms. Instead, it is a very strong arm, which corresponds to "力 (power)".

### C.3 Chinese Idiom Format

Chinese idioms are a special kind of words, which often have specific formats and semantic information, so they cannot directly translate the meaning of a single emoji and concatenate words into sentences. The most common format is four-character idioms, which often come from ancient Chinese myths, historical stories, classics, etc., consisting of four Chinese characters, with a Chinese literary style, and often a symmetrical structure. In addition, multi-character idioms, although far fewer in number than four-character idioms, are equally important components. Some of them have less than four words (e.g., three-character idioms) and some have more than four words. Generally speaking, whether it is a four-character idiom or a multi-character idiom, it follows the one-to-one relationship between emoji and characters, but there are special cases.

**Chinese character mapping**  Usually, idioms follow a one-to-one relationship between emojis and characters, but there are special cases. First of all, there will be multiple emojis corresponding to one character. Often, many numbers will have this correspondence, especially those with large digits. For instance, " 1 0 0 0 0 " denotes the "万" (ten thousand). In addition, there is a mapping relationship between multiple characters in an emoji. This kind of correspondence is relatively rare, usually in multi-character idioms, and this one-to-many mapping relationship occurs when two or more characters can form a new word represented by an emoji. The above two mapping relationships require MLLM to further complete the understanding and reasoning of multiple emoji contexts on the basis of recognizing the meaning of a single emoji.

## C.4 ENGLISH WORD TASK

**English Word Split**   In the English word task, unlike the regular one word corresponding to one emoji, it is common for multiple emoji to represent one word. The task usually splits the word, corresponds multiple emoji to different parts, and finally synthesizes them into one word. For example, "blackBerry" is split into "black-", "ber-", "-ry", and then the 🐻 is utilized to represent "ber-", and finally, the box of black and the letter "E" is added to get "blackBerry". The word "Panda" can be split into "Pan-" and "-da," where "Pan-" corresponds to 🔍. This kind of word splitting usually does not occur alone but is also accompanied by the linguistic phenomenon of harmonic words with many-to-one mapping. For example, in the word "lemon", the word is split into "le-" and "-mon", then "mon-" is harmonized as "man", and 👨 is chosen to represent the split syllable "mon". Beyond understanding the meaning of individual emojis, the MLLM must also remove unnecessary letters and combine the parts to infer a completely new word.

## C.5 ENGLISH IDIOM TASK

**Harmonization Word**   Similar to Chinese idioms, there are also a lot of harmonic characters in English idioms. Sometimes difficult to find directly related emoji to represent, harmonic characters with similar pronunciations are often chosen to replace them. For example, "To be loaded", "To" harmonizes with "Two" 2️⃣, and "be" harmonizes with "bee" 🐝. Most English idioms still keep the simple direct correspondence between emoji and words. What is more challenging for English idioms is their Abstract visual comprehension and word mapping reasoning problem.

**Abstract visual Emoji Understanding**   In English, for emoji that cannot be represented by direct correspondence, the data do not tend to choose harmonic words, but further associate related emoji, putting further demands on the reasoning ability of MLLM. For example, in "As genuine as a three-dollar bill", "genius" is usually accompanied by intellect and inspiration, and so a shining star ✨ is used to represent the image of sparkling inspiration of such genius. This deeper level of image comprehension requires a greater understanding of the meaning of the image and the text behind it.

**English Word Mapping**   Unlike most one-to-one relationships in Chinese idioms, there are a large number of non-one-to-one correspondences in English idioms. Due to the large number of articles, prepositions, conjunctions, and other words in English that are difficult to directly use emojis, such words are usually omitted in the emoji representation of English idiom, and only the most critical nouns, adjectives, verbs, etc., are retained to express the core meaning. Therefore, the prediction process of English idiom is not a one-to-one translation mapping, which also poses more challenges to the ability of MLLM. For example, in "An apple a day keeps the doctor away.", for MLLM, it is necessary to reason out such common idioms just for the emojis of 🍎 and 🏥. This examines the internal knowledge-mining ability of the large language model and the strong reasoning ability. However, this kind of reasoning is also very easy to cause the hallucination problem.

# D   ADDITIONAL DETAILS OF EVALUATION METRICS

Since our primary goal is to propose the emoji-to-idiom task and assess MLLM's ability to understand and reason about the textual semantics corresponding to abstract visual information, our work primarily focuses on task formulation, data construction, and the underlying assessment approach. We believe this task fills a crucial gap in evaluating MLLM's visual capabilities in representing abstract symbols and bridging the visual-verbal divide. Therefore, our current assessment metrics compare predicted answers with standardized answers that have undergone rigorous automated and manual filtering across multiple granularities.

When we calculate the word-level metrics, we need to match the correct answers exactly, and here we also include the consideration of structural information. The accuracy between the output response and the ground truth of the character-level model does not take into

account the structural one-to-one correspondence, but rather divides and acquires the answer by character, and calculates it at the character level, as long as the character level can be matched with the ground truth, it can be regarded as a correct character.

## D.1 Overview of the Design of Metrics and How to Use

**Word-level (in Chinese idiom and English word) / Sentence-level (in English idiom):** This is the most direct measure of MLLM's ability to fully understand the semantic information of the symbols in the image. When MLLM's output and the standard answer can be matched exactly at word-level or sentence-level (including structural matches), i.e., when MLLM successfully outputs the correct complete idiom, MLLM is considered to have answered the question correctly. At this level, we computed the associated precision, recall, and F-1 values. At this point, MLLM possesses both the understanding of individual emoji, and moreover the corresponding reasoning ability and text generation ability, which is the one that satisfies our initial motivation and truly realizes the ability of unified visual-linguistic understanding. Therefore, this is the most direct indicator of MLLM's ability.

**Character-level (in Chinese idiom and English word)/Word-level (in English idiom):** due to the greater challenge of this benchmark, we found that without additional training, it is more difficult for MLLM to fully answer the correct and complete idiom. In order to better analyze which part of emoji-to-idiom comprehension is more challenging for MLLM, we evaluated at character-level/word-level and calculated Precision, recall, and F-1 values. Specifically, for English idiom, we computed BLEU-1 vs. BLEU-2 to better measure MLLM correctness at this level. Since we did not consider structural information in this segment, the Character/word-level metrics reflect more on MLLM's ability to understand individual emoji, due to which there are still a large number of emoji that just need to understand their meanings directly without additional reasoning. Therefore, MLLM's ability to understand the emoji themselves is reflected when MLLM receives a higher score in this item. If MLLM's score in the first item slips very significantly compared to the second item, we can conclude that MLLM possesses basic emoji comprehension skills but lacks further reasoning skills.

**Semantic similarity:** After we computed the character/word-level with exploring Chain-of-thought reasoning, we could not help but notice that sometimes MLLM is actually better at understanding individual emoji, predicting one or two characters correctly, but performs poorly at the full idiomorphic level. but poorer performance on the complete idiom level. There are even some MLLMs that correctly determine the meaning of each emoji during the CoT process, but when outputting the idiom, they output an idiom that has similar semantics but is completely different at the character level, resulting in serious semantic drift or even hallucination. Therefore, we added an extra step of calculating the metrics for the semantic similarity of the output response to the standard answer.

- We use LLM (specifically GPT-4o) as an expert to score the semantic similarity of the output response to the standard answer. The specific scoring criteria are as follows: scoring is done on a scale of 1-5, with 1 being completely dissimilar, 2 being relatively dissimilarity, 3 fairly similar, 4 being relatively similar, and 5 being completely similar. The specific prompt we use for scoring is: "Please measure the semantic similarity between the given standard answer and the model output on a scale of 1 to 5, where 1 means completely dissimilar, 2 means relatively dissimilarity, 3 means fairly similar, 4 means relatively similar, and 5 means completely similar. Output only a numerical score."

- The semantic similarity metrics can be complemented with Character-level/word-level metrics, both of which play an important role when the MLLM is unable to fully match the standard answer at the idiomorphic level. The semantic similarity metric focuses more on whether the answers output by the model are semantically similar to the standard answers, and does not focus on the understanding of individual emoji, but rather reflects an overall comprehension of the semantics of the text directly from the images.

We provide a detailed description of the evaluation metrics and the different capabilities of MLLM they embody in the emoji2idiom, which helps researchers to use our benchmark and assess the specific capability bottlenecks of MLLM.

## D.2 Details of Evaluation Metrics for different tasks

### D.2.1 Chinese Idiom Task

In the task of Chinese idioms, we evaluate them separately at the word level and at the character level. At the word level, we first calculate the Word level accuracy, which is the ratio of the number of words that exactly match the ground truth to the total number of words. In order to further validate the image-to-language comprehension and reasoning ability of MLLM, we further propose the Chr-1 and Chr-2 indicators at the word level, which represent the ratio of the number of words with one or more characters correctly and two or more characters correctly compared to ground truth, to the total number of words. At the character level, we compare the difference between each character in the predicted word and each character in the ground truth to calculate the Precision, Recall, and F-1 values.

### D.2.2 English Word Task

In the task of English words, we evaluate them separately at the word level and at the character level. At both levels, we compare the difference between the predicted word/character and each word/character in the ground truth, calculating the Precision, Recall, and F-1 values.

### D.2.3 English Idiom Task

In the task of English idioms, we evaluate them separately at the sentence level and at the word level. At both levels, we compare the difference between the predicted sentence/word and each sentence/word in the ground truth, calculating the Precision, Recall, and F-1 values. In addition, to further measure the similarity of the generated sentences to ground truth, we further calculated BLEU-1 and BLEU-2 values.

## D.3 Disccusion of Metrics

Since our primary goal is to propose the emoji-to-idiom task and **assess MLLM's ability to understand and reason about the textual semantics corresponding to abstract visual information**, our work primarily focuses on task formulation, data construction, and the underlying assessment approach. We believe this task fills a crucial gap in evaluating MLLM's visual capabilities in representing abstract symbols and bridging the visual-verbal divide. Therefore, our current assessment metrics compare predicted answers with standardized answers that have undergone rigorous automated and manual filtering across multiple granularities, and we have not yet explored further metrics in our evaluation.

### D.3.1 Discussion about Ground Truth

It is worth noting that an emoji has different meanings in different cultures and contexts, which is one of the key challenges in emoji-to-idiom task. Therefore, instead of focusing on understanding the direct meaning of a **single** emoji (in fact, the current MLLM can directly give multiple possible meanings for a single emoji), we provide a specific contextual **context** (a sequence of multiple emojis with a specific semantic meaning) to limit the semantic of

single emoji. In addition, the correct answer of **emoji sequence** needs to meet the meaning of each emoji in the sequence and the structural information of the sequence, which largely avoids the generation of multiple possible answers.

Certainly, in the process of data collection, we did encounter a very small number of scenarios where other answers were barely acceptable. For example, "💪 = 💰", the standard answer is "Health is wealth", while the other possible answer is "Money is power".But there are two problems here: 1) the predicted answer does not fully satisfy the structural information of the sequence, i.e., translating the idiom from left to right.2) The length of this emoji sequence is very short, which makes the possible prediction results more variable. As the length of the sequence becomes longer, the less likely it is that other matching answers will appear.In our data, the average length of the series is 4.11, 4.23, 7.48, 5.32 in Chinese four-character idioms, English words, Chinese multi-character idioms, and English idioms. Therefore, we believe that it is feasible to provide a standard answer to predict the outcome for evaluation, and to measure the consistency of emoji and text.

### D.3.2 Discussion about Furthur Metrics

In future work, we plan to develop additional evaluation metrics to better assess MLLM's ability to bridge the multimodal divide between vision and language. Our goals include:

- **Adding semantic similarity metrics**: Our current metrics primarily quantify the direct correspondence between the standard answer and the generated result, with a relative lack of semantic similarity calculation. Incorporating semantic similarity into our evaluation, particularly in human assessments, will provide a more comprehensive measure of performance.

- **Measuring the similarity between the emoji's original visual information and the final prediction:** By annotating emojis with a standardized language base, we can compare results to predictions more effectively. For example, the emoji "☀️" might correspond to the textual interpretation "sun, 阳 (read as 'yang')" and relate to the harmonic word "养 (read as 'yang')" in the final ground truth. While a predicted result like "日 (sun)" might not match the direct character level, **it captures the initial visual information of the emoji and should be scored accordingly**. This step will help identify specific bottlenecks MLLM faces in this task, whether in visual understanding, harmonic character mapping, or textual reasoning.

- **Including GENERATION metrics:** In addition to common generative metrics (e.g., ROUGE, METEOR, diversity, complexity), we will consider task-specific metrics, such as adherence to idiomatic format specifications, like meeting the four-character idiom requirement.

## E  Additional Details of Baselines and Implementation details

### E.1 Baselines

We select close-source MLLMs, GPT-4V and GPT-4o, to evaluate the emoji2idiom benchmark.

**GPT-4V**  Building on the work done for GPT-4, GPT-4 with vision (GPT-4V) enables users to instruct GPT-4 to analyze image inputs provided by the user.

**GPT-4o**  GPT-4o ("o" for "omni") accepts as input any combination of text, audio, image, and video, which is similar to human response time(opens in a new window) in a conversation. In our work, we choose the GPT-4o-20240513 as our baseline.

To conduct a richer evaluation, we select a series of open-source MLLMs for testing, including Qwen-VL Bai et al. (2023), DeepSeek-VLLu et al. (2024a), LLaVa Li et al. (2023b), and CogAgent Hong et al. (2023).

**Qwen-VL-7B** Qwen-VL (Qwen Large Vision Language Model), proposed by Alibaba Cloud, accepts images, text, and bounding boxes as inputs. It provides Multi-lingual LVLM supporting text recognition and Abstract visual recognition and understanding.

**DeepSeek-VL-7B** DeepSeek-VL is an open-source MLLM designed for real-world vision and language understanding applications, which possesses general multimodal understanding capabilities.

**LLaVA-1.5-7B** LLaVA is a MLLM that connects a vision encoder and a language model for visual and language understanding, which uses instruction tuning data generated by GPT-4.

**CogAgent-18B** CogAgent-18B supports image understanding based on CogVLM, which further possesses GUI image Agent capabilities.

### E.2 Implementation Details

In our experiments, we explore the inference capabilities of MLLM to accomplish multiple tasks. In the GPT-4v and GPT-4o tests, we call the official API and use the original temperature coefficient for the experiment. The time of the GPT4v and GPT4o experiments in this work has been updated to May 30, 2024. It is important to note that since the closed-source model GPT series will be updated over time, the reproduction of results in future studies may be affected by the GPT version. In the experiments of the closed-source model, we use the original official weights for evaluation without additional training. For Qwen-VL, we use the open-source model of Qwen-VL-7B and experiment on a single NVIDIA RTX 3090. For DeepSeek-VL, we experiment with DeepSeek-VL-7B-chat on an NVIDIA RTX 3090. We implement CogAgent-18B on 2 NVIDIA RTX 3090 cards for FP16 inference, and LLaVA-1.5-7B is also implemented with 2 NVIDIA RTX 3090 cards. For all the evaluations, we set the temperature as 0.7 and top-k as 0.9. We further provide the computation source and time usage in Table 7. The `Emoji2Idiom` data and evaluation scripts can be found on GitHub https://anonymous.4open.science/r/Emoji2Idiom-0CCA.

### E.3 Prompt Template

#### E.3.1 General Prompt

For different MLLMs, the templates of the input prompt and message are naturally different due to the different ways the models were originally called. In the MLLM assessment, our prompt design mainly follows the following principles. (1) Keep it as short as possible. Provide effective information in a short prompt to avoid interfering with the understanding of MLLM. (2) Ensure the consistency of the prompts of different MLLMs as much as possible. This ensures that our evaluation results are not affected by the prompt. (3) The design of the different models is designed to give the task concerns more clearly. We show our prompt as shown in Figure 12.

#### E.3.2 CoT Prompt

We design the CoT process, inspired by human thinking when seeing the emoji2idiom task.

1. Understand each emoji and provide a directly related textual representation.

2. Generate possible harmonic words, fine-grained comprehension, and idiom associations.

3. Combine multiple emojis to ensure the idioms align or find other possible matches.

4. Finalize the text and check for grammatical errors.

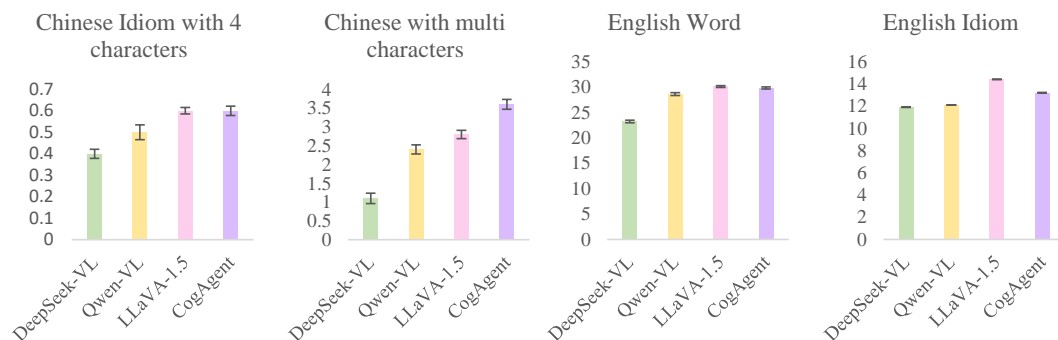

Figure 11: The error bar graphs of different evaluation results of MLLM, which illustrate the Word accuracy of Chinese idiom with four words and Multi-words, Word-level precision of English Word, and Sentence-level precision of English idiom task, respectively.

Table 7: The usage of the computation source and time of MLLMs.

| Model | Hardware | Time Usage | Model | Hardware | Time Usage |
|---|---|---|---|---|---|
| GPT-4v | API | 156min | DeepSeek-VL-7b | 1 RTX 3090 | 719min |
| GPT-4o | API | 149min | CogAgent-18b | 2 RTX 3090 | 503min |
| Qwen-VL-7b-chat | 1 RTX 3090 | 623min | LLaVA-1.5-7b | 2 RTX 3090 | 562min |

## F    ADDITIONAL DETAILS OF AUTOMATIC EVALUATION

To ensure the reliability and robustness of the results, we set up three different random seeds for the experiment in the automatic evaluation of the open-source model and take the average value as the final experimental result. The resulting error bar diagram is shown in Figure 11.

The detailed information of the total amount computed and the type of resources used is shown in Table 7.

## G    ADDITIONAL DETAILS OF HUMAN EVALUATION

### G.1    HUMAN EVALUATION GUIDELINE

We invite human experts to conduct human assessments, one for human performance on `Emoji2Idiom` and one for scoring MLLM results. The specific evaluation guideline is shown in the Figure 13.

### G.2    DETAILED EVALUATION RESULTS

Based on these evaluation guidelines, human experts were able to obtain results from the evaluation of human performance on the `Emoji2Idiom` and the evaluation results of MLLMs. The specific results are shown in the Table 8 and Table 9.

## H    ADDITIONAL DETAILS OF CASE STUDY

### H.1    TYPICAL CASE STUDY

**Harmonization Problem**    There are a large number of harmonic character phenomena in our dataset, which poses a great challenge to the understanding and reasoning of the large language model. The MLLM is also significantly hampered by these harmonic words during emoji understanding. As shown in the Fig. 15 🎀 stands for "捷" and 合 stands for "河", and the model does not succeed in recognizing any of these harmonic words. The

| Prompt Template for Evaluation of MLLMs |
|---|
| We provide the details of our prompt designed for different MLLMs for evaluation on different tasks in our Emoji2Idiom benchmark. |

| Task Definition |
|---|
| **Task:** Emoji-to-Chinese Idiom / Emoji-to-English Word / Emoji-to-English Idiom
**Identifier:** Chinese Idiom, English Word, English Idiom
**Output:** Chinese Idiom, English Word, English Idiom |

| Prompt Template | |
|---|---|
| ➢  **Without In-context learning and additional training, we evaluate the inference ability.** | |
| **Qwen-VL** | 'text': 'What is the \<Identifier> represented by the emojis in this image? Output format: The \<Output> is...',
'image': file_path |
| **DeepSeek-VL** | "content": "'What is the \<Identifier> represented by the emojis in this image? Output format: The \<Output> is...",
"images": ["file_path"] |
| **LLaVA-1.5** | 'text': 'What is the \<Identifier> represented by the emojis in this image? Output format: The \<Output> is...',
'image': file_path |
| **CogAgent** | 'text': 'What is the \<Identifier> represented by the emojis in this image? Output format: The \<Output> is...,
'image': file_path |
| **InternVL-2** | 'text': 'What is the \<Identifier> represented by the emojis in this image? Output format: The \<Output> is...,
'image': file_path |
| **GPT-4v/GPT-4o/Claude-3.5-sonnet** | {"type": "text",
"text": "What is the \<Identifier> represented by the emojis in this image? Output format: 'The \<Output> is...'."},
{ "type": "image_url",
 "image_url": {
  "url": f"data:image/jpeg;base64,{base64_image}"} |
| ➢  **Without additional training, we evaluate the inference ability and In-context learning.** | |
| **Qwen-VL** | 'text': 'What is the \<Identifier> represented by the emojis in this image? Output format: The \<Output> is...'
'image': file_path
'text': 'Here are some \<Task> examples of the emoji images and the corresponding idioms. Emojis come first, and follow the corresponding \<Identifier> .'
'image': example_image_1
'text': 'The idiom is \<ground truth> .' |
| **GPT-4o** | "text": "What is the \<Identifier> represented by the emojis in this image? Output format: 'The \<Output> is...'."
"image_url": {"url": f"data:image/jpeg;base64,{base64_image}"
"text": "Here are some \<Task>  examples of the emoji images and the corresponding idioms. Emojis come first, and follow the corresponding \<Identifier> ."
"image_url": {"url": f"data:image/jpeg;base64,{base64_example_image_1}"}
"text": "The idiom is \<ground truth>" |

Figure 12: Our prompt template is designed for evaluation on MLLMs.

| Guideline of Human Evaluation of MLLM Performance |
|---|
| This study aims to evaluate the quality of MLLM performance on our benchmark Emoji2Idiom. Each case provides you with task type, an emoji image, answer, and ground truth, You need to evaluate the generated answer from the following aspects. |

| Case | |
|---|---|
| **Task:** Emoji-to-Chinese Idiom
**Generated Answer:** 闻鸡起舞
**Ground Truth:** 捷雷不及掩耳 | **Image:** |

| Evaluation Metrics | |
|---|---|
| ➢ **Normality:** Whether the generated answer conforms to the idiom's specifications, including formatting specifications and semantic specifications | |
| **Options** | 1. Completely non-standard   2. Mostly non-standard   3. Fairly standard
4. Mostly standard   5. Completely standard |
| **Examples** | 1.  "朝三暮四" shows completely standard to the normality.
2.  "天鹅绒门盘" shows completely non-standard.
3.  "眼大眼小耳朵瞎" shows mostly non-standard. |
| ➢ **Semantic similarity**: Whether the generated answers are semantic similar to the ground truth | |
| **Options** | 1. Completely dissimilar   2. Mostly dissimilar   3. Fairly similar
4. Mostly similar   5. Completely similar |
| **Examples** | 1.  "眼见为实" shows completely dissimilar to the ground truth "星星点点".
2.  "杞人忧天" shows fairly similar to the ground truth "闷闷不乐". |
| ➢ **Emotional similarity:** Whether the generated answers are emotional similar to the ground truth | |
| **Options** | 1. Completely dissimilar   2. Mostly dissimilar   3. Fairly similar
4. Mostly similar   5. Completely similar |
| **Examples** | 1.  "狐假虎威" shows mostly similar to the ground truth "阴魂不散".
2.  "班门弄斧" shows mostly dissimilar to the ground truth "当务之急". |
| ➢ **Visual similarity:** Whether the generated answers are visually similar to the origin emoji image | |
| **Options** | 1. Completely dissimilar   2. Mostly dissimilar   3. Fairly similar
4. Mostly similar   5. Completely similar |
| **Examples** | 1.  "*Money is power*." is mostly similar to the origin emoji image
2.  "*bright idea*." is fairly similar to the emoji image |
| ➢ **Fluency:** Whether the generated answers are fluency and easy to understand. | |
| **Options** | 1. Completely influent   2. Mostly influent   3. Fairly fluent
4. Mostly fluent   5. Completely fluent |
| **Examples** | 1.  "*Break the ice*." is mostly fluent.
2.  "日日山如故" is completely influent. |
| ➢ **Complexity:** Whether this task is complex for the MLLM and thus difficult to solve. | |
| **Options** | 1. Completely easy   2. Mostly easy   3. Fairly complex
4. Mostly complex   5. Completely complex |
| **Examples** | 1.  " ⭐⭐👐 corresponds to 星星点点 " is mostly easy to solve.
2.  " 👵👀📖✏ corresponds to 先睹为快 " is mostly complex to solve. |

Figure 13: The human evaluation guideline.

| Problems | English Word | | English idiom | |
|---|---|---|---|---|
| **Couldn't identity** *homophonic* *characters* (Red color denotes the homophonic characters or sound-like characters) | Emoji: 🗝️➕Ⓦ

Ground truth: kiwi

Qwen-VL: keyplus
DeepSeek-VL: W
LLaVA:key word
CogAgent:key word
GPT-4V: keywest
GPT-4o:keyword | Emoji: Ⓞ➕🚶➕Ⓖ

Ground truth: Orange

Qwen-VL: OG
DeepSeek-VL: go
LLaVA: go
CogAgent: on the go
GPT-4V:Jog
GPT-4o:ONGOING | Emoji: 🐋 I hope so!

Ground truth: Well, I hope so

Qwen-VL: I hope so!
DeepSeek-VL: I hope so!
LLaVA: I hope so!
CogAgent: Whale, I hope so!
GPT-4V:whale of a time
GPT-4o:Whale, I hope so | Emoji: 2️⃣🪲💼

Ground truth:To be loaded

Qwen-VL: busy as a bee
DeepSeek-VL: busy as a bee
LLaVA: busy as a bee
CogAgent:GPT-4V:busy as a bee
GPT-4o:busy as a bee |
| Suffer *hallucination* problem | Emoji: ⚔️🐋

Ground truth: Killer whale

Qwen-VL: sea danger
DeepSeek-VL:swordfish
LLaVA: whale
CogAgent: swordfish
GPT-4V: swordfish
GPT-4o: swordfish | Emoji: 👨🦅

Ground truth: Bald eagle

Qwen-VL: eagleman
DeepSeek-VL: eagle-eyed
LLaVA: eagle
CogAgent: bald eagle
GPT-4V: headphones
GPT-4o: eagle | Emoji: 🐴⬆️

Ground truth: To pony up

Qwen-VL: horse
DeepSeek-VL: rising to the occasion
LLaVA: horse
CogAgent: horse up
GPT-4V: straight from the horse's mouth
GPT-4o: horsing around | Emoji: 🚶🎅📇💰

Ground truth: To go from rags to riches

Qwen-VL: to wear someone's shirt
DeepSeek-VL: keeping one's shirt on.
LLaVA: Bring home the bacon
CogAgent: pay for some money
GPT-4V: A man after my own heart
GPT-4o: walk away from a deal |
| *One emoji to multi-character* mapping or vice versa | Emoji: ⬛➕🐻➕Ⓔ

Ground truth: Blackberry

Qwen-VL: black bear
DeepSeek-VL: BEAR
LLaVA: black bear
CogAgent: bear
GPT-4V: Bare
GPT-4o: Squarebear | Emoji: ⬜🦍

Ground truth: Grape

Qwen-VL: gray gorilla
DeepSeek-VL: square gorilla
LLaVA: gorilla
CogAgent: gorilla
GPT-4V: gorilla
GPT-4o: Kong | Emoji: 👉❄️

Ground truth: Blow off steam

Qwen-VL: plugging away
DeepSeek-VL: blowing in the wind
LLaVA: blow off
CogAgent: blow the snow away
GPT-4V: cold shoulder
GPT-4o: blow hot and cold | Emoji: 🔔

Ground truth: Ring a bell

Qwen-VL: to ring a bell
DeepSeek-VL: to ring the bell
LLaVA: ring a bell
CogAgent: a bell
GPT-4V: sound the alarm
GPT-4o: saved by the bell |
| *Fine-grained image* *understanding* of the emoji symbol | Emoji: 🌍🐘

Ground truth: African elephant

Qwen-VL: Earth elephant
DeepSeek-VL: elephant
LLaVA: elephant
CogAgent: elephant on earth
GPT-4V: worldwide
GPT-4o: elephant | Emoji: 🐱🏛️

Ground truth: Caterpillar

Qwen-VL: cat green pillar
DeepSeek-VL: cat
LLaVA: cat
CogAgent: cat
GPT-4V: catastrophe
GPT-4o: cathedral | Emoji: 😊🐟🧍😄👴😔🎂

Ground truth: It is never too old to learn.

Qwen-VL: fly by the seat of your pants
DeepSeek-VL: grinning from ear to ear.
LLaVA: Thinking helps a lot.
CogAgent: Practice makes perfect.
GPT-4V: an emotional rollercoaster
GPT-4o: To err is human; to forgive, divine. | Emoji: 📦🦵

Ground truth: Receive a kickback

Qwen-VL: think outside the box
DeepSeek-VL:open the box and find money inside
LLaVA: a box of money
CogAgent: box outside
GPT-4V: think outside the box
GPT-4o: out of the box |

Figure 14: Four typical problems the MLLM suffer in English word and idiom tasks.

| Problems | Chinese idiom with four character | | Chinese idiom with more than four characters | |
|---|---|---|---|---|
| Couldn't identity *homophonic characters* (Red color denotes the homophonic characters or sound-like characters) | Emoji:

Ground truth: 难舍难离

Qwen-VL: 虎视蛇行
DeepSeek-VL:家徒四壁
LLaVA: 杯弓蛇影
CogAgent: 蛇蝎心肠
GPT-4V: 南瓜蛇果
GPT-4o: 南瓜蛇离 | Emoji:

Ground truth: 玉宇琼楼

Qwen-VL:金碧辉煌
DeepSeek-VL: 高楼大厦
LLaVA:一举两得
CogAgent: 醉生梦死
GPT-4V:黄粱美梦
GPT-4o:五谷丰登 | Emoji:

Ground truth: 捷雷不及掩耳

Qwen-VL:电闪雷鸣
DeepSeek-VL: 耳聪目明
LLaVA:拔鸡代猴
CogAgent:束手无策
GPT-4V:闻鸡起舞
GPT-4o:闻鸡起舞 | Emoji:

Ground truth: 河水不犯井水

Qwen-VL:哭笑不得
DeepSeek-VL:一见钟情
LLaVA:泥菩萨过江
CogAgent:滴水穿石
GPT-4V:一波三折
GPT-4o:泪流满面 |
| Suffer *hallucination* problem | Emoji:

Ground truth: 以肉喂虎

Qwen-VL:狗急跳墙
DeepSeek-VL:卧虎藏龙
LLaVA:虎口拔牙
CogAgent:狼心狗肺
GPT-4V:狐假虎威
GPT-4o:如坐针毡 | Emoji:

Ground truth: 抱火厝薪

Qwen-VL:喜新厌旧
DeepSeek-VL:火光冲天
LLaVA:旧的不去新的不来
CogAgent:笑而不答
GPT-4V:新官上任三把火
GPT-4o:笑里藏刀 | Emoji:

Ground truth: 不问三七二十一

Qwen-VL: 三七二十一
DeepSeek-VL:问鼎中原
LLaVA:女大十八变
CogAgent:三七二十一
GPT-4V:三心二意
GPT-4o:颠三倒四 | Emoji:

Ground truth:项庄舞剑，意在沛公

Qwen-VL:背井离乡
DeepSeek-VL: 盲人摸象
LLaVA:投笔从戎
CogAgent:望洋兴叹
GPT-4V:珠光宝气
GPT-4o:草木皆兵，风声鹤唳 |
| *Multi-emoji to one* character mapping | Emoji:

Ground truth:万壑千岩

Qwen-VL:一丝不苟
DeepSeek-VL:一声不吭
LLaVA:一鸣惊人
CogAgent:一见钟情
GPT-4V:一心一意
GPT-4o:差强人意 | Emoji:

Ground truth: 晴空万里

Qwen-VL: 四大皆空
DeepSeek-VL: 艳阳高照
LLaVA: 空谷幽兰
CogAgent: 空空如也
GPT-4V:一暴十寒
GPT-4o:晴天霹雳 | Emoji:

Ground truth: 中河失舟，一壶千金

Qwen-VL: 一举成名
DeepSeek-VL:画龙点睛
LLaVA:一言九鼎
CogAgent: 狡兔三窟
GPT-4V:朝三暮四
GPT-4o:狐假虎威 | Emoji:

Ground truth: 朝朝寒食，夜夜元宵

Qwen-VL: 冰冻三尺非一日之寒
DeepSeek-VL: 有口难言
LLaVA:日出而作，日入而息
CogAgent:一日三秋
GPT-4V:破釜沉舟
GPT-4o:日月潭 |
| *Fine-grained image understanding* of the emoji symbol | Emoji:

Ground truth: 精才绝艳

Qwen-VL: 鸟语花香
DeepSeek-VL: 掌上明珠
LLaVA: 海阔天空
CogAgent: 鲸吞蚕食
GPT-4V:东施效颦
GPT-4o:鲤鱼跃龙门 | Emoji:

Ground truth: 抵足谈心

Qwen-VL: 心神不宁
DeepSeek-VL: 心悦诚服
LLaVA: 以退为进
CogAgent: 心服口服
GPT-4V: 下落不明
GPT-4o: 兵贵神速 | Emoji:

Ground truth: 知无不言，言无不听

Qwen-VL: 无所不能
DeepSeek-VL: 笑口常开
LLaVA: 对牛弹琴
CogAgent: 对牛弹琴
GPT-4V:人山人海
GPT-4o:见不得人 | Emoji:

Ground truth:止谤莫若自修

Qwen-VL: 十年寒窗
DeepSeek-VL:一波未平，一波又起
LLaVA: 一个巴掌拍不响
CogAgent: 一问三不知
GPT-4V:瓜田李下
GPT-4o:杯弓蛇影 |

Figure 15: Four typical problems the MLLMs suffer in Chinese idiom tasks.

Table 8: Human performance on Chinese idiom task.

| Human performance | word-level | character-level | | Complexity | Time usage |
|---|---|---|---|---|---|
| | | character-2 | character-1 | | |
| Human expert-1 | 56 | 65 | 76 | 3.5 | 15s per image |
| Human expert-2 | 69 | 74 | 81 | 3.1 | 22s per image |
| Human expert-3 | 64 | 70 | 85 | 3.2 | 28s per image |
| Human expert-4 | 77 | 85 | 95 | 3.4 | 24s per image |
| Average | 66.5 | 73.5 | 84.25 | 3.3 | 22s per image |

Table 9: Human evaluation on Chinese idiom task.

| Model | Expert | Chinese idiom | | | | | English idiom | | | | |
|---|---|---|---|---|---|---|---|---|---|---|---|
| | | Std. | Sem. | Emj. | Emo. | Flu. | Std. | Sem. | Emj. | Emo. | Flu. |
| GPT-4v | 1 | 3.7 | 1.1 | 1.3 | 2.4 | 3.7 | 4.4 | 2.3 | 2.5 | 2.1 | 4.3 |
| | 2 | 4.5 | 1.6 | 1.8 | 2.6 | 3.9 | 4.2 | 2.1 | 2.4 | 2.2 | 4.5 |
| | 3 | 3.9 | 1.1 | 1.4 | 2.1 | 3.6 | 4.3 | 2.2 | 2.5 | 2.3 | 4.4 |
| | 4 | 3.9 | 1.2 | 2.4 | 2.2 | 3.8 | 4.3 | 2.2 | 2.7 | 2.4 | 4.5 |
| | Avg. | 4.0 | 1.3 | 1.7 | 2.3 | 3.8 | 4.3 | 2.2 | 2.5 | 2.3 | 4.4 |
| GPT-4o | 1 | 4.1 | 1.4 | 1.8 | 2.3 | 3.8 | 4.4 | 2.2 | 2.6 | 2.5 | 4.4 |
| | 2 | 4.8 | 1.6 | 2.1 | 2.3 | 4.0 | 4.3 | 2.2 | 2.6 | 2.3 | 4.7 |
| | 3 | 4.1 | 1.4 | 1.5 | 2.4 | 3.6 | 4.5 | 2.3 | 2.4 | 2.2 | 4.2 |
| | 4 | 4.5 | 1.7 | 2.5 | 2.3 | 3.9 | 4.2 | 2.4 | 2.6 | 2.5 | 4.5 |
| | Avg. | 4.1 | 1.4 | 1.8 | 2.3 | 3.8 | 4.4 | 2.3 | 2.6 | 2.4 | 4.5 |

inference of such harmonic words requires the help of relevant contexts, and in our data scenario, the model is required not to understand individual emoji alone, but to deeply and comprehensively analyze the context of the emoji. Obviously, under this task requirement, current multimodal large language models are not well equipped to capture emoji context with harmonic word reasoning.

**Hallucination Problem**    During the process of recognizing emoji, the model can usually recognize the corresponding meaning of individual emoji better. At this point, the models are prone to hallucinations. After recognizing the meaning of a single emoji, they think diffusely about this emoji and only consider words or idioms directly related to the emoji, ignoring the involvement of emoji in other contexts. For example, in Fig. 15 the model recognizes 🐎 and starts thinking about idioms related to horse and directly outputs "horsing around" without considering another emoji. Similarly, when MLLMs capture 🔥, they search for the Chinese idiom with the character "火 (fire) ". Another example shows that the GPT-4v and GPT-4o recognize the number 3 and directly associate it with the idiom "朝三暮四" and "颠三倒四", which contains the number 3, without considering the information of the rest of the emoji around.

**Multi-to-One or One-to-Multi Character Mapping.**    For the MLLM, it is customary to perform a one-to-one mapping operation where an emoji corresponds to a Chinese character or English word. In many scenarios, however, it is necessary to perform a multi-to-one or one-to-multi mapping. For example, in Figure 15 the number 1 0 0 0 is composed of four emojis, but the model does not successfully combine them into one character "千 (one thousand)". And in Figure 14, 🔔 not just indicates a single "bell" or "alarm", but the idiom "ring a bell". This reasoning relies on the capability of knowledge ming and the reasoning based on the emojis in images and their corresponding linguistic meanings.

**Abstract visual Image Understanding of the Emoji Symbol**    The model shows good performance in simply recognizing the shallow meanings of individual emoji, but in Abstract visual understanding, it is difficult to match the emoji information with the context to get

the deep corresponding emoji meanings. For example, in Figure 14, the prediction of the idiom "receive a kickback", the model simply captures the emoji 📦, the meaning of "box", and interprets it as "think outside of the box" or "out of the box", but does not combine the package attributes of "receiving something" with the hint of money to generate the correct answer.

# I  ADDITIONAL DETAILS OF TRAINING AND FINETUNING FOR FUTURE WORK

Based on these results and error case studies, we propose potential training methods and frameworks that could significantly improve MLLM performance in visual-linguistic tasks, drawing inspiration from human approaches to joint visual-semantic reasoning:

- Direct fine-tuning: We can incrementally pre-train MLLMs on an emoji-rich corpus to build a basic understanding of emoji. Our initial tests indicate that MLLMs already demonstrate a foundational grasp of emoji, performing well in many cases. Following pre-training, we suggest a 4:1 division of the fine-tuning dataset and test set, with direct fine-tuning on the pre-trained MLLM. This method mirrors human learning, where repeated practice after initial knowledge acquisition leads to mastery in a specific domain.

- Incorporating Chain of Thought (CoT) design: When translating emoji to idioms, we can model the process after human reasoning. This CoT design references the process of human thinking and reasoning, which can assist MLLM to think about idiom generation in a structured way, and is better able to further analyze where exactly MLLM goes wrong and provide inspiration for subsequent research work. We hope that such reasoning can be further generalized to more general symbol understanding, and our emoji2idiom data can also be used as part of general symbol understanding to evaluate the general symbol understanding capability of the large language model.

- Adding a symbol mapping set as external knowledge: A single emoji may correspond to multiple characters. By constructing an emoji-to-character mapping set, we can enable MLLM to learn possible alignments. This approach is similar to how humans use external knowledge to accomplish tasks that might be challenging without it.

- Multi-agent invocation: Referring to the CoT process, we can utilize multiple intelligences for tasks like emoji comprehension, harmonic word association, and emoji combination, allowing for integrated task planning, memory iteration, and refined reasoning.

Finally, our work significantly contributes to enhancing the visual comprehension and reasoning capabilities of MLLMs. Most current unified evaluation metrics focus on MLLM's understanding of natural images, often overlooking abstract visual information and symbolic representations—areas that receive less attention during training. Additionally, MLLMs struggle with recognizing complex textual information in images, particularly handwritten text or intricate symbols. We believe our emoji2idiom task not only complements existing evaluations of abstract symbolic representations but also offers a solution for deeper visual reasoning, thus promoting the development of visual-textual alignment and multimodal unification architecture.

