# OpenReview forum: "🤔Emoji2Idiom: Benchmarking Cryptic Symbol Understanding of Multimodal Large Language Models"
_ICLR.cc/2025/Conference — ICLR 2025 Conference Withdrawn Submission_

### Official Review · Reviewer_BCKJ · 2024-10-23

**Soundness:** 3
**Presentation:** 3
**Contribution:** 2
**Rating:** 6
**Confidence:** 4

**Summary:**

The paper presents the Emoji2Idiom benchmark aimed at evaluating the cryptic symbol understanding capabilities of Multimodal Large Language Models (MLLMs). It introduces a novel task where MLLMs must translate emojis in images to corresponding idioms, challenging models to understand semantic correlations and reason linguistic meanings from visual data. The authors construct a high-quality benchmark with tasks spanning language, text format, and semantic diversity. Experiments with advanced MLLMs demonstrate current models' limitations in understanding visual data's linguistic information, suggesting areas for future research.

**Strengths:**

* The paper introduces a unique benchmark task that bridges vision and language modalities, providing a new perspective on MLLM evaluation.
* The proposed Emoji2Idiom benchmark covers diverse language tasks, enhancing the generality of the evaluation.
* The whole work is complete, with detailed annotations and filtering, and diverse evaluation settings such as zero-shot, few-shot, COT, and human evaluation.

**Weaknesses:**

* Why is it necessary for emoji as image task compared to emoji as text task? There needs a more in-depth discussion about the relationship between emoji as image task and high-level vision-language understanding.
* For some emojis in Emoji2Idiom benchmark, there may not be a standard answer (the same emoji can mean different things and emotions in different situations, where everyone has their own perspective). How do the authors ensure that the ground truth answer is unique?
* The authors need to provide more latest MLLMs to support experimental conclusions, such as Claude-3.5-Sonnet,  LLaVA-1.6, Qwen2-VL, and InternVL2.
* Table 3 has a format error about "Deepseek-VL".

**Questions:**

* Have the authors considered designing multiple-choice questions where the model selects the correct answer from given options rather than generating the answer? It is worth exploring whether there is a significant gap compared to the generation task.
* The paper could improve by including tests on model performance after fine-tuning with data highly relevant to the task of emoji understanding. This would provide a clearer picture of how adaptable and improvable the models are with targeted training.
* The dataset's reliance on manual annotation makes it difficult to expand, which could limit its long-term applicability as models improve.

I look forward to an active discussion with the authors during the rebuttal phase and will revise my score accordingly.

**Details Of Ethics Concerns:**

The paper describes the source of the dataset in detail, so I don't have any ethical concerns.

---

> ### Author Response · Authors · 2024-11-21
> **Response to reviewer BCKJ (Part 1/3)**
>
> We are very grateful for your recognition and interest in our work, and your insightful comments are very valuable in enhancing our work. We also look forward to discussing further with you about our emoji2idiom.
> > Weakness 1: Why is it necessary for emoji as image task compared to emoji as text task?
>
> **Response**
>
> We apologize for causing you confusion about the reason for emoji as image input and our motivation of unified vision-language benchmark. We hope our response can address your concerns.
>
> - **For image input**: we agree that when people use emoji as input, it's usually in the form of text alongside other text. However, the emoji itself is a **visual symbol**. When understanding emoji, from the perspective of **human brain cognition**, people in real scenarios do not understand emoji directly as “text characters”, but rather **understand the visual information** of emoji from a visual perspective, so as to obtain the specific linguistic meaning represented by emoji. It makes sense to use emoji as image input to test the comprehension ability of LLM.
> - **For motivation of abstract symbol as high-level vision-language task**: the main motivation of this benchmark is **not only** to focus on the use of LLMs that can understand emoji, but also to choose **an abstract symbol** such as emoji, which combines **both visual representations and text-specific semantic information**, to achieve a unified understanding of vision and language. That is, as we mentioned in the paper, to directly understand the semantic information of cryptographic symbols in images. This benchmark is different from previous benchmarks based on **VQA** forms, which represent visual and textual information **separately**, but directly condenses the features of **both into** the abstract symbols understanding of emoji. Therefore, we chose **the image form of emoji input**.
> - **For real-world scenario**:  in real-life scenarios, people sometimes utilize emoji instead of textual characters for conversations, and although the input is in textual format at this point, what **LLM** really needs to have is **visual understanding**. With our proposed data, LLMs can be better challenged and trained to understand abstract symbols, which is more helpful to understand complex emoji representations of real-life scenarios.
>
> We have provided more details about the image input and benchmark motivation to eliminate potential confusion for readers, please note that you can see the lateset PDF version we uploaded especially the Page 1, Line 19-20; Page 2, Line 93-95; Page 3, Line123-128 in blue font.
>
> > Weakness 2: For some emojis in Emoji2Idiom benchmark, there may not be a standard answer (the same emoji can mean different things and emotions in different situations, where everyone has their own perspective). How do the authors ensure that the ground truth answer is unique?
>
> **Response**
>
> We apologize for making it confused to understand the how we obtain the unique ground truth. As we have already mentioned in Sec 3.3 (Page 6, Line 288-294) and Appendix D.4 (Page 20, Line 1040-1057) about the influence of **different cultural backgrounds and different emoji semantics** on the standard answer.
>
> - First, it is true that for a **single** emoji, it may correspond to different meanings. However, when an emoji is placed in a **specific semantic** context, its meaning becomes more certain. We have created an **“emoji sequence”** environment to avoid such answer uncertainty.
> - For a complete emoji sequence, although there are multiple possible meanings for a single emoji, the answer can be almost uniquely determined within **a sequence due to the semantics of the emoji in context** (i.e., in neighboring locations) as well as the **overall structure** of the emoji (sequential emojis tend to be output one-to-one in a sequential manner). For example, "💪 = 💰️", the standard answer is "Health is wealth", while the other possible answer is "Money is power". However, the predicted answer does not fully satisfy the structural information of the sequence, i.e., translating the idiom from left to right.
> - Of course, in such a case, there may still be a very small number of emoji sequences that may still have a non-unique answer under the constraints of contextual semantics and structural information, at which point we have **filtered the possible non-unique answers** based on human experts to ensure that the emoji sequences in our final data correspond to only one standard answer.

---

> ### Author Response · Authors · 2024-11-21
> **Response to reviewer BCKJ (Part 2/3)**
>
> > Weakness 3: The authors need to provide more latest MLLMs to support experimental conclusions, such as Claude-3.5-Sonnet, LLaVA-1.6, Qwen2-VL, and InternVL2.
>
> **Response**
>
> Thank you for your suggestion. The article mentions our considerations about the baseline of the assessment, and we chose the current model out of consideration for abstract symbolic visual reasoning and Chinese comprehension. Of course, we can indeed supplement the evaluation with more advanced models. Due to the time limit, we are selecting the closed-source model **Claude-3.5-Sonnet-20241020** and the open-source model **InternVL2-8B** that you mentioned to supplement the experiments, and updating the new experimental results to our latest version PDF Sec.4.2, as shown below. Please see the latest version of our PDF in openreview website， especially in Page 6, Line 278-279 and Page 7, Line 332-333. We are willing to **provide more experiment** in the furthur version.
>
> ### Idiom with Four Words (Word-level and Character-level)
>
> |     Model   | Word   | Chr-2 | Chr-1 | Pre. | Rec. | F-1 |
> |-------------|--------|-------|-------|------|------|-----|
> | InternVL-2  | 0.8    | 6.3   | 37.8  | 9.1  | 9.1  | 9.1 |
> | Claude-3.5  | 1.3    | 6.7   | 23.3  | 8.0  | 8.0  | 8.0 |
>
> ### Idiom with Multi-words (Word-level and Character-level)
>
> |     Model   | Word   | Chr-2 | Chr-1 | Pre. | Rec. | F-1 |
> |-------------|--------|-------|-------|------|------|-----|
> | InternVL-2  | 3.4    | 8.3   | 29.4  | 8.9  | 15.6 | 11.3|
> | Claude-3.5  | 1.4    | 2.9   | 7.1   | 6.0  | 9.7  | 7.4 |
>
> ### English Word (Word-level and Character-level)
>
> |     Model   | Word   | Chr-2 | Chr-1 | Pre. | Rec. | F-1 |
> |-------------|--------|-------|-------|------|------|-----|
> | InternVL-2  |  31.1  | 31.1 | 31.1 | 56.6  | 57.2 | 56.9 |
> | Claude-3.5  | 42.3  | 42.3 | 42.3 | 63.9  | 73.8 | 68.5  |
>
> ### English Idiom (Sentence-level and word-level)
>
> |     Model   | Word   | Chr-2 | Chr-1 | Pre. | Rec. | F-1 | B-1 | B-2 |
> |-------------|--------|-------|-------|------|------|-----|-----|-----|
> | InternVL-2  |  15.3  | 15.3  | 15.3  | 19.3 | 22.1 | 20.6 | 18.4 | 16.1 |
> | Claude-3.5  |  29.8  | 29.8  | 29.8  | 48.0 | 42.7 | 45.2 | 42.3 | 39.7 |
>
> > Weakness 4: Table 3 has a format error about "Deepseek-VL".
>
> **Response**
>
> We apologize for our writing mistakes and appreciate your corrections. We have revised this section and submitted the latest version of the PDF (Page 7,Line 329) on openreview.

---

> ### Author Response · Authors · 2024-11-21
> **Response to reviewer BCKJ (Part 3/3)**
>
> > Question 1: Have the authors considered designing multiple-choice questions where the model selects the correct answer from given options rather than generating the answer? It is worth exploring whether there is a significant gap compared to the generation task.
>
> **Response**
>
> Thank you very much for your insightful suggestion. We did consider the multiple-choice question format during the data construction process, and also made some attempts. However, during our experiments as well as reviewing related studies, we found that the multiple-choice question format of the benchmark is prone to **bias**, which requires more methods for eliminating bias [1] [2]. It is very likely that the large language model did **not really answer the question correctly**, but only **predicted an option** as an answer, which can greatly affect the **validity** of the benchmark and produce experimental results with bias. In addition, the **order in which the options** are set can also affect the **robustness** of the assessment [3], and the model will be very sensitive to changes in the position of different options when predicting the results. All of these factors can affect the validity of the assessment when evaluated, and therefore, we ultimately did not choose multiple choice questions for our test problems.
>
> [1] Li W, Li L, Xiang T, et al. Can multiple-choice questions really be useful in detecting the abilities of LLMs?[J]. arXiv preprint arXiv:2403.17752, 2024.
>
> [2] Myrzakhan A, Bsharat S M, Shen Z. Open-LLM-Leaderboard: From Multi-choice to Open-style Questions for LLMs Evaluation, Benchmark, and Arena[J]. arXiv preprint arXiv:2406.07545, 2024.
>
> [3] Zheng C, Zhou H, Meng F, et al. Large language models are not robust multiple choice selectors[C]//The Twelfth International Conference on Learning Representations. 2023.
>
> > Question 2: The paper could improve by including tests on model performance after fine-tuning with data highly relevant to the task of emoji understanding. This would provide a clearer picture of how adaptable and improvable the models are with targeted training.
>
> **Response**
>
> Your advice on fine-tuning and training is very valuable to the entire MLLM community. However, since our motivation of this work is to **propose such a task of unifying visual-linguistic understanding task with the benchmark**, we provide in our experiments an exploration of the model's raw reasoning ability (Sec. 4.2: Page 6, Line 319-323, Page 7, Line 347-367), ICL ability (Sec. 4.3: Page 7, Line 371-377, Page 8, Line 402-404), and CoT (Sec. 4.3: Page 8, Line 406-413) ability. We do not go further with the training and fine-tuning experiment, as we think that including an analysis of training was **not necessary** in a task that focuses primarily on dataset and benchmark.
>
> In addition, in Appendix I (Page 28, Line 1482-1511, Page 29, Line 1512-1517) of the paper, we also propose an **exploration of further training and fine-tuning strategies** to help provide a clearer picture of how adaptable and improvable the models are with targeted training on emoji data. We will be exploring this in subsequent work.
>
> > Question 3: The dataset's reliance on manual annotation makes it difficult to expand, which could limit its long-term applicability as models improve.
>
> **Response**
>
> **For current version of emoji2idiom:** In this work, we did rely on manual filtering to ensure **high quality data**, and thus construct a small but high-quality benchmark to evaluate the performance of MLLM, which is sufficiently large compared to other benchmarks.
>
> **For extension of data:** As the model develops and is further explored, the need for quantity scale will be higher. We have provided a pipeline for **automatic data generation pipeline** in the text (Page 4, Figure 2, and Sec. 3.2), which includes web search and text-to-emoji generation that can be used for further generation. If we want to reduce the labor cost, the **manual filtering process can be done by LLMs**, which will be trained and fine-tuned using the **high-quality data** we already have. In addition, we can use these high-quality emoji-idiom sequences as seed data based on the **self-instruct method** to utilize MLLM for relevant data generation (which can utilize our existing text-to-emoji generation method).

---

> > ### Comment · Reviewer_BCKJ · 2024-11-21
> > **Respond to the responses to Paper 6776**
> >
> > Thanks for the further explanation and clarification of your response. I will raise my rating to 6 according to the discussion.

---

> ### Author Response · Authors · 2024-11-22
> **Response to the response to Reviewer BCKJ**
>
> Thank you for your recognition of our work and your efforts during the review phase, and your valuable comments are very helpful in improving the quality of our paper. We would be happy to discuss this with you further!

---

### Official Review · Reviewer_eqny · 2024-10-27

**Soundness:** 2
**Presentation:** 3
**Contribution:** 2
**Rating:** 3
**Confidence:** 3

**Summary:**

This paper introduces Emoji2Idiom, a benchmark for evaluating multimodal large language models' ability to translate emoji sequences into meaningful text (Chinese idioms, English words, and English idioms).

 The benchmark challenges models in three key aspects: harmonized word reasoning, fine-grained visual understanding, and complex mapping relationships. Through extensive evaluation of commercial and open-source MLLMs, the authors demonstrate current limitations in models' ability to bridge visual and linguistic understanding. The work's main contributions are proposing a novel emoji-to-text translation task and providing insights for improving MLLMs' symbolic understanding capabilities.

**Strengths:**

Using emoji-to-text translation as a proxy for testing abstract symbolic understanding is both original and insightful, especially given emojis' unique position as symbols with both visual and linguistic meaning.

The work meaningfully addresses a gap in current MLLM evaluation benchmarks - while most focus on natural image understanding, this paper tackles the understudied area of abstract symbol comprehension, providing a reusable benchmark and practical insights for improving model performance.

**Weaknesses:**

The paper's primary weakness lies in its artificial data construction approach. The extensive use of homophonic relationships to create emoji sequences, feels contrived and deviates significantly from how emojis are naturally used in real-world communication. In reality, people use emojis to express emotions, enhance communication efficiency, or circumvent content restrictions - not to create puzzles through sound-alike words. This artificial construction, especially in the text-to-emoji generated portion of the dataset, makes many examples challenging even for humans to understand and raises questions about the benchmark's real-world applicability. A more meaningful evaluation would focus on how MLLMs understand emoji usage in authentic social media contexts, where emoji sequences naturally emerge from genuine communication needs rather than being reverse-engineered from predetermined phrases.

**Questions:**

Is there overlap between the human experts in data construction and the human experts in model evaluation?

---

> ### Author Response · Authors · 2024-11-21
> **Response to Reviewer eqny**
>
> Thank you for your comments and the effort you put into the review phase. We hope our response can address your concerns.
>
> > Weaknesse 1: The paper's primary weakness lies in its artificial data construction approach. In reality, people use emojis to express emotions, not to create puzzles through sound-alike words. This artificial construction, especially in the text-to-emoji generated portion of the dataset, makes many examples challenging even for humans to understand and raises questions about the benchmark's real-world applicability.
>
> **Response**
> We note that your main concern is the applicative gap between choosing a task model like emoji2idiom and the real world, and the gap between applying it to the real world by manually constructing the data. We hope our response can help to address your concerns.
>
> - **For task form of Puzzle**: When people use emoji, they usually express emotions or **replace text as communication**, e.g. we want to say “sunny” and we use ☀ to replace.
>   - At this time, they usually use harmonic characters to replace the expression. We chose the form of Puzzle not because we just want MLLM to realize the understanding of emoji Puzzle, **but to measure MLLM's ability to understand this kind of abstract visual symbols, which is helpful to understand the usage of emojis to replace textual expression**.
>   - In addition, we do not only focus on MLLM's understanding of emoji , but we **regard emoji as a special kind of abstract symbols that combines both visual representations and textual semantics**, can be used as a representative to realize a **unified multimodal comprehension ability** that is different from the previou VQA based benchmark that treat the vision and language seperately. It will also be helpful for **other symbols understanding in the real world (e.g., chemical structure formulas, musical notation)**.
>
> - **For artificial data construction**: As we describe in detail the motivation and operation of our data construction in Sec. 3.2 (Page 4, Line 196-206), in our data construction:
>   - A very large portion of the data is **obtained from the Internet and real social media**, which are reasonable emoji-idoim pairs that people recognize as fitting their own usage habits.
>   - The reason for artificially constructing a text-to-emoji corpus is to **alleviate the problem of data bias**, since the data obtained directly from real social media contains a large number of **repetitive emoji-idoim mappings**, and this data bias can easily make the **LLM hallucination**. Therefore, we need to additionally construct some other possible emoji-text correspondences different from these repeated mappings.
>   - We also mention in Sec.3.2 (Page 5, Line 228-230) that in order to avoid such construction results being too strange and **not in line with the language habits** people use, we **remove** the emoji-text pairs that do not match the human usage habits in human filtering to enhance their applicability in the real world.
>
> > Questions : Is there overlap between the human experts in data construction and the human experts in model evaluation?
>
> **Response**
>
> We select different human experts who are native speakers with extensive experience and give a detailed guideline to mitigate subjectivity. So there is **no overlap** in human construction and human evaluation.

---

> > ### Comment · Reviewer_eqny · 2024-11-21
> > **Response**
> >
> > First of all, thank you for your reply. I agree with what you said about the means to improve the quality of the dataset. However, after reviewing the data, I think that you did not screen the data so strictly in practice. This may be my bias as a Chinese-English bilingual, so I will lower the confidence, but will not  raise rating to accept.

---

> > > ### Author Response · Authors · 2024-11-22
> > > **Response to Review eqny for data filtering (Part 2/3)**
> > >
> > > ### Human Filtering：
> > >
> > > In this phase, we invited human experts in Chinese and English languages to perform manual data filtering, which included the following steps in total:
> > >
> > > - **Duplicate value checking**: for the automatic filtering phase, the machine flags a portion of emoji-idiom pairs where the text is the same but the corresponding images are not the same. the human expert needs to further check whether the emoji expressions here are really different. For the pairs with identical emoji images, the human expert will delete them.
> > >
> > > - **Image quality check**: Human experts further check whether the emoji images are unclear and illegible, and remove the illegible images.
> > >
> > > - **Idiom standardization check**: Human experts need to check whether the idiom text expression is standardized, including the format of the idiom, whether it has a specific linguistic meaning, and whether it is in line with common human usage, etc., to ensure that our dataset meets the real-world usability. For example, for the idiom “blue sky and white clouds”, although it is a four-word idiom that conforms to the norms of human usage, it does not have a specific allusion, mythological story, traditional story background, or special semantic meaning, and does not belong to the standard idioms. For example, although “流水高山” is a four-letter word with a specific historical background, people more often use the expression “high mountains and flowing water”. Therefore, “流水高山” is not an expression that conforms to human language usage and will be deleted.
> > >
> > > - **Emoji and Idiom Relevance Check**: Since in emoji to idiom expression, many times the representation of harmonic characters will be utilized, which will increase the difficulty of emoji comprehension and the difficulty of generating the final idioms. Human experts will evaluate the relevance of emoji to idioms:
> > >   - If too many or too complex harmonic characters are used with the emoji representation, at this time the task will be too difficult for not only MLLM but also humans to understand. At this point, the human expert will consider the relevance of this emoji sequence to the idiom to be too low and delete the emoji-idiom pair.
> > >   - It is noteworthy that we conducted an evaluation of human ability on this benchmark in Sec. 4.4 (Page 8, Line 424-431, Page 9, Line 448-452) and found that humans achieved an average score of 66.5 on the word-level accuracy of Chinese idioms. This score demonstrates both that our dataset is challenging and that the task is accomplishable, and that there is still much room for improvement in the performance of the current MLLM on this task.
> > >   - In addition, we are considering adding a distinction between different comprehension difficulties and different comprehension domains to the version2 version of the emoji2idiom dataset, which would further improve the validity of our evaluation.
> > >
> > > - **Repeated harmonic word mapping check**: due to the limited expression of emoji, when using emoji to replace textual expressions, harmonic words are often used to find the corresponding emoji for expression. emoji2idiom also has a large number of harmonic words. However, we found that if the mapping of the same emoji corresponding to a certain harmonic word occurs too many times, it may cause data bias to LLM in subsequent training, i.e., when LLM sees this emoji it automatically thinks of this harmonic word that occurs multiple times. To mitigate the bias caused by this harmonic word mapping, we performed:
> > >   - Count the repeated emoji-character harmonic word mappings, and when there are more than ten occurrences, we manually replace the expression of the emoji (find other harmonic word counterparts to replace the original repeated emoji), or just delete the redundant emoji-idiom pair.
> > >   - In addition, we also considered this issue during the original data collection. Our retrieval and collection in different sources of the original emoji database can reduce this duplicate mapping. We also take different generation methods when manually constructing text-to-emoji data, which also helps to increase the diversity of harmonic word mappings.
> > >
> > > - **Safety and Ethics Check**: Based on the automatic detection, the human experts further conducted a safety and ethics check of the emoji images and idiom text, checking whether there are any issues such as violent gore, abusive language, sexism, racial discrimination, stereotyping, and so on, in the data.
> > >
> > > In order to **mitigate the subjectivity** of the human experts and at the same time enhance the validity of the human assessment criteria, we provide a **detailed guideline** for the human experts, which contains an explanation of each assessment criterion, relevant examples, and quantitative scoring criteria. This guideline can be seen in detail in **Appendix A, Figures 7 and 8 (Page 15 and 16)** in our PDF.

---

> > > ### Author Response · Authors · 2024-11-22
> > > **Response to Review eqny for data filtering (Part 3/3)**
> > >
> > > After manual screening, the statistics of our Stage 2 data are shown in the table below:
> > >
> > > |  Task   |  Before Human-filtering |   |      | After Human-Filtering |   |        |
> > > |---------|-------|-------|------------------|-------|-------|-------------------|
> > > |         |Image-Idiom pair  | Idiom | Image |Image-Idiom Pair  | Idiom | Image  |
> > > |Chinese idiom (4 character) |  2252 | 2252  | 2252  |  1876 | 1876  | 1876  |
> > > |Chinese idiom (multi-character)|  576 | 576  | 576  |  334 | 334  | 334   |
> > > |English Word |  1076  | 1076   | 1076  |  842  | 842   | 842  |
> > > |English Idiom|  1182 | 1182 | 1182  |  783 | 783 | 783  |
> > >
> > > As a result, we believe that we have strictly filtered our data.

---

> > > > ### Author Response · Authors · 2024-12-01
> > > > **Friendly Follow-Up on Rebuttal and Response**
> > > >
> > > > As the rebuttal phase nears its end, thank you for your response and the positive discussions. We are glad to hear your recognition of our work and are eager to address any remaining concerns.
> > > >
> > > > In response to your continued questions about our data filtering process, we have provided more detailed explanations, supplemented with additional statistical figures, and incorporated these updates into Appendix A.1 of the PDF. We hope that this additional information will help clarify our data filtering procedures and demonstrate how we have constructed a high-quality dataset.
> > > >
> > > > We sincerely hope to resolve your concerns and gain your endorsement, as it greatly motivates us to continue our research and strive to improve our dataset. We understand your time is extremely valuable, but with the deadline approaching, we still genuinely hope to address your concerns and look forward to your feedback.

---

> > > > ### Author Response · Authors · 2024-12-02
> > > > **Friendly Follow-Up on Discussion about Data Filtering**
> > > >
> > > > Thank you for your time and effort on our manuscript, we are glad to hear your recognition of our work and are eager to address any remaining concerns about our **data filtering** process. We have provided more **detailed explanations, supplemented with additional statistical figures, and incorporated these updates into Appendix A.1** of the PDF. We hope that this additional information will help clarify our data filtering procedures and demonstrate how we have constructed a high-quality dataset.
> > > >
> > > > As the rebuttal phase nears its end, we sincerely hope to resolve your concerns and gain your endorsement, as it greatly motivates us to continue our research and strive to improve our dataset! look forward to your feedback.

---

> ### Author Response · Authors · 2024-11-22
> **Response to Review eqny for data filtering (Part 1/3)**
>
> Thank you for your response. Noting that you have concerns about our data filtering process, we further explain the specific steps of our data filtering and the related motivations, which can help us maintain the high quality of our data in both Chinese and English contexts. We hope that our response addresses your concerns about our data filtering.
>
> ### Automatic Filtering：
>
> In this phase, we mainly utilize machines and large language models to filter large-scale data, which includes the following steps in total:
>
> - **Deletion of default values**. We utilize a machine to automatically remove incomplete emoji-idiom pairs, including those with missing corresponding emoji images and those with missing corresponding idioms. It is guaranteed that each emoji image corresponds to the standard idiom answer one by one.
>
> - **Removing Duplicate Values**. We utilize the machine to automatically remove duplicate emoji-idiom pairs.Here, we only need to remove the emoji-idiom pairs corresponding to identical emoji sequences while retaining the pairs with the same idiom text but corresponding to different emoji representations, which helps to enhance the diversity of the dataset. Note that we will first filter the pairs corresponding to the same idiom text by the machine with additional labels, and make a manual decision on whether to perform the deletion in the next stage of manual filtering.
>
> - **Image Quality Check**. We utilize LLM (specifically GPT-4o is used), to perform image quality checking, which entails marking and removing: images that are too blurry and those that do not meet the ethical norms (images that contain elements of violence, abusive language, discrimination, etc.) along with their corresponding idioms. e.g., for the emoji sequence 🥣2️⃣🙅🏻🅱, which hides elements of verbal abuse, it does not comply with the security ethics and therefore will be removed.
>
> - **Text Ethics Checking**. We utilize LLM (specifically GPT-4o) to perform text ethics checking, which involves tagging and deleting idiom with elements of violence, discrimination, abuse, etc. For example, “红颜祸水” is a sexist idiom, and we will delete its corresponding emoji-idiom pair.
>
> After automated filtering, our first stage data statistics are shown in the table below:
>
> |  Task   |  Before Auto-filtering |   |      | After Auto-Filtering |   |        |
> |---------|-------|-------|------------------|-------|-------|-------------------|
> |         |Image-Idiom pair  | Idiom | Image |Image-Idiom Pair  | Idiom | Image  |
> |Chinese idiom (4 character) |  2338    |  2379  |2362  |  2252 | 2252  | 2252   |
> |Chinese idiom (multi-character)|  622    |  641  |629  |  576 | 576  | 576   |
> |English Word |  1261   |  1289  |1263  |  1076  | 1076   | 1076  |
> |English Idiom|  1237    |  1254  |  1244  |  1182 | 1182 | 1182  |

---

> ### Author Response · Authors · 2024-11-24
> **Looking Forward to Further Discussion and Your Insights about Data Filtering**
>
> Dear Reviewer eqny,
>
> Thank you for your valuable feedback and for highlighting concerns about our data filtering process. Upon reviewing your comments, we have provided detailed explanations of our data filtering methods, including the specific procedures and criteria applied, as well as relevant statistical data to support the process. We hope these additional details address your concerns and offer a clearer understanding of how we ensure data quality.
>
> As the ICLR discussion phase is nearing its conclusion, we are eager to engage in further communication with you. Your expertise and insights would greatly help us refine our work and further enhance the quality of our benchmark. We look forward to your thoughts and would deeply appreciate your feedback at your earliest convenience. Thank you once again for your time and thoughtful contributions to this review process.
>
> Best regards,
>
> The Authors of Manuscript 6776

---

> > ### Author Response · Authors · 2024-11-26
> >
> > We sincerely appreciate the time and effort you’ve dedicated to reviewing our work, as well as your recognition of its contributions. Your feedback has been invaluable in refining our research, and we are truly grateful for your support.
> >
> > We understand that you still have questions regarding our data filtering approach. While we have provided further detailed explanations, we have not yet received a follow-up response from you. If there are additional issues or concerns, we would be happy to address them. Additionally, while the discussion phase has been extended, the deadline for submitting a revised PDF remains unchanged. This presents some challenges for further discussion and revisions. We hope to hear back from you at your earliest convenience to discuss any unresolved issues.
> >
> > Thank you once again for your selfless assistance and thoughtful feedback.

---

### Official Review · Reviewer_hb6G · 2024-11-01

**Soundness:** 1
**Presentation:** 2
**Contribution:** 2
**Rating:** 3
**Confidence:** 5

**Summary:**

This paper introduces a novel benchmark focused on translating emojis in images into corresponding idioms or phrases using multimodal large language models (MLLMs). The concept is innovative and intriguing, with experiments conducted to evaluate the performance of several MLLMs on this benchmark.

**Strengths:**

1. The paper presents a new perspective on evaluating MLLMs' understanding capabilities, with a creative and interesting approach.

**Weaknesses:**

1. The primary data source was the internet, but it seems the authors did not consider potential data leakage during filtering;
2. Some evaluation settings are vague and need further explanation. For example, why is there a difference in templates used across models? Does the metric calculation consider language structure, and if not, why exclude semantics? How are model responses processed?
3. The authors claim that Emoji2Idiom is a fine-grained visual understanding task, but the task design does not reflect this. There is no granular categorization of tasks from a broader perspective; perhaps describing it as an abstract understanding capability would be more appropriate;
4. The range of evaluated models is limited, and neither open-source nor closed-source models specify which versions were used, with open-source models not even providing parameter sizes.

**Questions:**

1. Many tasks in Emoji2Idiom seem inclined toward a matching task. Did the authors consider testing the performance of CLIP models or using them as a data filtering method?
2. Since emojis have multiple visual representations, how did the authors address this factor? Were there any ablation studies conducted?
3. Did the authors consider annotating the dataset with difficulty, domain, etc., to provide more analytical insights?
4. The authors mention using GPT-4 to review images during data filtering. This might be a typo error; it may be necessary to clarify which specific model was used (GPT-4V, GPT-4o, or ChatGPT web).

---

> ### Author Response · Authors · 2024-11-21
> **Response to reviewer hb6G (Part 1/3)**
>
> We appreciate your comments and the effort you put into the review phase. We hope our response explains your confusion and addresses your concerns about emoji2idiom.
>
> > Weakness 1: The primary data source was the internet, but it seems the authors did not consider potential data leakage during filtering;
>
> **Response**
>
> Thanks for your comment. When we collect the raw data, we refer to the previous work such as [1][2][3][4]. Their raw image data are **obtained from the Internet**, and we believe that this scheme of retrieving from the Web as a means of evaluating the raw data is feasible, and that this data sourced from the real Internet enhances the **applicability of our Benchmark in the real world**. In addition, to the best of our knowledge, emoji2idiom is a **novel task**, and while past LLM training data will contain training corpus of emoji, there is **no task with related data for emoji to idiom.**
>
> [1] Hong Y, Zhen H, Chen P, et al. 3d-llm: Injecting the 3d world into large language models[J]. Advances in Neural Information Processing Systems, 2023, 36: 20482-20494.
>
> [2] Li B, Ge Y, Ge Y, et al. SEED-Bench: Benchmarking Multimodal Large Language Models[C]//Proceedings of the IEEE/CVF Conference on Computer Vision and Pattern Recognition. 2024: 13299-13308.
>
> [3] Fu, Chaoyou et al. “MME: A Comprehensive Evaluation Benchmark for Multimodal Large Language Models.” ArXiv abs/2306.13394 (2023): n. pag.
>
> [4] Ge Z, Huang H, Zhou M, et al. Worldgpt: Empowering llm as multimodal world model[C]//Proceedings of the 32nd ACM International Conference on Multimedia. 2024: 7346-7355.
>
> > Weakness 2: Some evaluation settings are vague and need further explanation. For example, why is there a difference in templates used across models? Does the metric calculation consider language structure, and if not, why exclude semantics? How are model responses processed?
>
> **Response**
>
> We sincerely apologize for the confusion caused by our description of the evaluation phase. We provide a detailed description of our evaluation phase in terms of **prompt template design, metrics computation, processing of model responses, and add experiments for semantic similarity evaluation**. These additions and experiments have been updated in the latest version of the PDF, Appendix E.3.1 (Page 21, Line 1130-1133, Page 22, Line 1157-1170), Sec.3.4 (Page 6, Line 298-303) and Appendix D (Page 19, Line 990-1025, Page 20, Line 1026-1028), Sec.4.2 (Page 7, Line 337-345 and Line 360-367). We hope this addresses your concerns.
>
> - **Template**: For different MLLMs, the templates of the input prompt and message are naturally different due to the **different ways** the models were originally called. For example, an interaction in the form of chat is used when calling GPT-4v and GPT-4o, while an instruction type prompt is used when calling deepseek. In addition, the prompt templates for **processing image inputs** are also somewhat different for different models, which is the reason why our templates are not exactly the same. However, **to ensure a fairer comparison, we design our prompts in such a way that their prompts are as consistent as possible**, except for minor differences in formatting. We provide a description of our full Prompt in Appendix E.3.1 Page 21, Line 1130-1133, Page 22, Line 1157-1170, **and have also corrected two errors in Figure.12 (Page 12)** in this new version.
>
> - **Metric Calculation and response process**: The calculation of our evaluation metrics has been described in detail in Sec.3.4 and Appendix D (Page 19, Line 990-1025, Page 20, Line 1026-1028) in the text. When we calculate the **word-level** metrics, we need to match the correct answers exactly, and here we also include the **consideration of structural information**. The accuracy between the output response and the ground truth of the **character-level** model does **not take into account the structural** one-to-one correspondence, but rather divides and acquires the answer by character, and calculates it at the character level, as long as the character level can be matched with the ground truth, it can be regarded as a correct character. For example, if the ground truth is “虎虎生风” and the response is “三人成虎”, the character level of “虎” can be matched with the standard answer, and it can be scored.

---

> ### Author Response · Authors · 2024-11-21
> **Response to reviewer hb6G (Part 2/3)**
>
> > Weakness 2: Some evaluation settings are vague and need further explanation.
>
> **Response**
>
> - **Semantics similarity**: Since the main **motivation** of this paper is to propose emoji2idom, which is different from the **VQA-based MLLM benchmark** in the past, and use emoji, which is an abstract symbol that combines visual representation and special textual semantics, as a representative to realize a unified visual-semantic comprehension assessment. Therefore, we have chosen a **basic evaluation approach to compute pre.rec.F1.**. When evaluating the performance of the model, and of course, we have mentioned our expansion for further semantic comprehension evaluation in the detailed introduction to metrics in Appendix D.4 (Page 20, Line 1029-1076). Here we **add further experiments for evaluating semantic comprehension** by scoring the semantic similarity between response and groundtruth. Where the higher the similarity, the better the performance of the output answer is considered, which is more efficient than the computation without considering structural information, as shown in the following experimental results, and upload the result in Sec.4.2 Page 7, Line 337-345 and Line 360-367 in blue font. **Due to time limit**, we choose open-source MLLM InternVL-2 and closed-source MLLM GPT-4o to evaluate. We are willing to add more experiment in the final version.
>
> ### Chinese idiom with four words
>
> |              | Average Semantics | Distribution(%)|                 |                 |                 |                 |
> |--------------|------------------|-----------------|-----------------|-----------------|-----------------|-----------------|
> | Model        |                  | 1               | 2               | 3               | 4               | 5               |
> | InternVL-2   | 1.41             | 66.9            | 26.3            | 6.1             | 0               | 0.7             |
> | GPT-4o       | 1.66             | 56.7            | 29.1            | 9.5             | 0.6             | 4.1             |
>
> ### Chinese idiom with multi-words
>
> | Model        | Average Semantics | Distribution(%)|                 |                 |                 |                 |
> |--------------|------------------|-----------------|-----------------|-----------------|-----------------|-----------------|
> |              |                  | 1               | 2               | 3               | 4               | 5               |
> | InternVL-2   | 1.47             | 61.4            | 32.9            | 4.5             | 0               | 1.1             |
> | GPT-4o       | 1.76             | 59.1            | 25.0            | 5.7             | 1.2             | 9.0             |
>
> ### English word
>
> | Model        | Average Semantics | Distribution(%)|                 |                 |                 |                 |
> |--------------|------------------|-----------------|-----------------|-----------------|-----------------|-----------------|
> |              |                  | 1               | 2               | 3               | 4               | 5               |
> | InternVL-2   | 2.75             | 46.2            | 11.5            | 19.2            | 17.2            | 40.4            |
> | GPT-4o       | 3.55             | 27.5            | 7.8             | 2.0             | 7.5             | 55.2            |
>
> ### English idiom
>
> | Model        | Average Semantics | Distribution(%)|                 |                 |                 |                 |
> |--------------|------------------|-----------------|-----------------|-----------------|-----------------|-----------------|
> |              |                  | 1               | 2               | 3               | 4               | 5               |
> | InternVL-2   | 2.75             | 60.3            | 19.0            | 11.1            | 4.8             | 4.9             |
> | GPT-4o       | 2.99             | 28.5            | 22.0            | 7.7             | 5.5             | 36.3            |
>
> > Weakness 3: The authors claim that Emoji2Idiom is a fine-grained visual understanding task, but the task design does not reflect this. There is no granular categorization of tasks from a broader perspective; perhaps describing it as an abstract understanding capability would be more appropriate;
>
> **Response**
>
> Thank you very much for your valuable suggestions, which can make the readability of our paper better. We **agree** that it would be more appropriate to consider emoji2idiom as an abstract visual comprehension task, as we don't have a finer-grained categorization of the levels. We have **revised this expression** in the updated version according to your suggestions. Please note that you can see the updated version we uploaded on the OpenReview website, especially Page 3, Line 112-113, Page 6, Line 283, Page 10, Line 524 and 536 in blue font.

---

> ### Author Response · Authors · 2024-11-21
> **Response to reviewer hb6G (Part 3/3)**
>
> > Weakness 4: The range of evaluated models is limited, and neither open-source nor closed-source models specify which versions were used, with open-source models not even providing parameter sizes.
>
> **Response**
>
> We sincerely apologize that we did not clearly provide the details of the evaluation model in the text. Thanks for your correction, in the **lateset PDF Sec. 4.1 Page 6, Line 310-314,** we **provide the version information and parameter information**of the MLLM of our experiment as follows: claude-3.5-sonnet-20241022, gpt-4-vision-preview, gpt-4o-20240513 , Qwen-VL-7B, DeepSeek-VL-7B, LLaVa-1.5-7B, CogAgent-18B, InternVL-2-8B.
>
> For the evaluated models, we choose the models in the paper for the consideration of MLLM's abstract visual reasoning ability and Chinese language comprehension. Considering more advanced models, we **supplement our experiments with the open-source model InternVL-8B and the closed-source model Claude-3.5-sonnet**:
>
> ### Idiom with Four Words (Word-level and Character-level)
>
> |     Model   | Word   | Chr-2 | Chr-1 | Pre. | Rec. | F-1 |
> |-------------|--------|-------|-------|------|------|-----|
> | InternVL-2  | 0.8    | 6.3   | 37.8  | 9.1  | 9.1  | 9.1 |
> | Claude-3.5  | 1.3    | 6.7   | 23.3  | 8.0  | 8.0  | 8.0 |
>
> ### Idiom with Multi-words (Word-level and Character-level)
>
> |     Model   | Word   | Chr-2 | Chr-1 | Pre. | Rec. | F-1 |
> |-------------|--------|-------|-------|------|------|-----|
> | InternVL-2  | 3.4    | 8.3   | 29.4  | 8.9  | 15.6 | 11.3|
> | Claude-3.5  | 1.4    | 2.9   | 7.1   | 6.0  | 9.7  | 7.4 |
>
> ### English Word (Word-level and Character-level)
>
> |     Model   | Word   | Chr-2 | Chr-1 | Pre. | Rec. | F-1 |
> |-------------|--------|-------|-------|------|------|-----|
> | InternVL-2  |  31.1  | 31.1 | 31.1 | 56.6  | 57.2 | 56.9 |
> | Claude-3.5  | 42.3  | 42.3 | 42.3 | 63.9  | 73.8 | 68.5  |
>
> ### English Idiom (Sentence-level and word-level)
>
> |     Model   | Word   | Chr-2 | Chr-1 | Pre. | Rec. | F-1 | B-1 | B-2 |
> |-------------|--------|-------|-------|------|------|-----|-----|-----|
> | InternVL-2  |  15.3  | 15.3  | 15.3  | 19.3 | 22.1 | 20.6 | 18.4 | 16.1 |
> | Claude-3.5  |  29.8  | 29.8  | 29.8  | 48.0 | 42.7 | 45.2 | 42.3 | 39.7 |
>
> > Question 1: Many tasks in Emoji2Idiom seem inclined toward a matching task. Did the authors consider testing the performance of CLIP models or using them as a data filtering method?
>
> **Response**
>
> Although our task requires the implementation of emoji to text mapping, it is not a matching task but closer to a **generative task**. Understanding the meaning of a single emoji-to-text mapping is only one part of the task; what is more important is to generate idoim with specific semantics based on multiple possible meanings as well as contextually relevant and structural information.
>
> In addition, our main motivation for this benchmark is that we wish to **evaluate the unified representation capability of MLLM** to understand text semantics directly at the visual level, and CLIP is not in the scope of MLLM and therefore was not measured.
>
> > Question 2: Since emojis have multiple visual representations, how did the authors address this factor? Were there any ablation studies conducted?
>
> **Response**
>
> For a particular emoji, it will have **different visual representations on different displays** (cell phones, PCs, etc.) and in **different systems** (Windows, Android, Apple). In order to standardize our data representation and fair evaluation, we use the **same visual representation** (PC representation icon under Windows) for all emoji collected, while retaining its corresponding **general UTF-8 encoded representation** to avoid different visual representations affecting the robustness of the evaluation.
>
> > Question 3: Did the authors consider annotating the dataset with difficulty, domain, etc., to provide more analytical insights?
>
> **Response**
>
> Thank you for your suggestion. We **agree** that providing a fine-grained difficulty or domain annotation for our data would indeed allow for better rubrics and analysis. However, in our work, we have focused more on **propose such an innovative work and exploring related assessments**, and therefore do not have more fine-grained labeling and classification. Due to the time limit, we haven't conduct fine-grained annotations, and **we are willing to consider expanding the data in ver2** to include more granular labeling of difficulty and domain.
>
> > Question 4: The authors mention using GPT-4 to review images during data filtering. This might be a typo error; it may be necessary to clarify which specific model was used (GPT-4V, GPT-4o, or ChatGPT web).
>
> **Response**
>
> Thank you for your correction. We used GPT-4o when filtering the images, and **we've made changes in the new version of the PDF (Page 4, Line 214)** to make this clearer.

---

> ### Author Response · Authors · 2024-11-24
> **Seeking Your Valuable Response on Our Rebuttal**
>
> Dear Reviewer Hb6G,
>
> Thank you for your thoughtful feedback and constructive suggestions. In response to your concerns regarding our data sources, experimental parameters, and evaluation details, we have provided detailed explanations and conducted additional evaluation experiments to strengthen our work.  We have also refined the writing based on your suggestions to improve clarity and readability. The updated PDF version of the manuscript has been submitted, with all changes highlighted in blue to facilitate your review.
>
> As the ICLR discussion phase is nearing its conclusion, we regret to note that we have not yet received your response to our rebuttal. We are eager to engage in further dialogue with you, as your expertise and perspective would significantly contribute to improving the quality of our paper. We would be most grateful if you could share your thoughts at your earliest convenience. Thank you once again for your valuable contributions to this review process.
>
> Best regards,
>
> The Authors of Manuscript 6776

---

> > ### Comment · Reviewer_hb6G · 2024-11-24
> >
> > Thank you for your response and efforts to address the issues. However, I still have the following key concerns:
> >
> > 1. SEED Bench, one of the listed references, demonstrates data leakage issues, as shown by MMstar[1], with similar problems also identified in works like LIME[2]. This indicates that collecting data from the internet carries inherent risks of data leakage. Additionally, the relevance of the data construction methods in 3D-LLM and WorldGPT to building a benchmark similar to this one is unclear.
> >
> >     [1] https://arxiv.org/abs/2403.20330
> >     [2] https://arxiv.org/abs/2409.06851
> >
> > 2. Ensuring fairness in evaluation is a fundamental aspect of any official benchmark. I assume the inconsistency in Fig 12 might just be a typographical error. Additionally, based on your writing style, it seems you should include the prompts used for the newly added models, but they are absent. I strongly suggest you maintain consistency throughout the paper to ensure fairness.
> >
> > 3. It is unclear which of the existing evaluation methods best reflects the models' actual performance on the benchmark. Providing a concise overview would be helpful. The newly added semantic similarity method, which uses LLM as a judge, aligns with common practices, but the specific LLM, prompts, and scoring criteria are not disclosed. This omission limits the practical utility of the results. A clear and universal scoring method, along with an explanation of the significance of each evaluation score, is essential for researchers to effectively use the benchmark.
> >
> > I will maintain my current score. Additionally, I suggest reviewing related benchmark papers, including those cited in your work. Their experimental designs may provide valuable insights for improving your study.

---

> > > ### Author Response · Authors · 2024-11-25
> > > **Response to Reviewer hb6G (Part 2/3)**
> > >
> > > Since the main motivation for our benchmark is not to present a benchmark that addresses data leakage, we did not discuss much about data leakage in our data construction. However, with reference to your list of methods for mitigating data leakage in the LIME and MMStar, we believe that our data **adequately meets these requirements** and is a high-quality dataset that minimizes the data leakage problem as much as possible. We add our statistics after automated and manual filtering to verify the effectiveness of our data filtering.
> > >
> > > |  Task   |  Before Filtering |   |      | After Auto-Filtering |   |        |After Human-Filtering |   |        |
> > > |---------|-------|-------|------------------|-------|-------|-------------------|-------|-------|-------------------|
> > > |         |Image-Idiom pair  | Idiom | Image |Image-Idiom Pair  | Idiom | Image  |Image-Idiom Pair  | Idiom | Image  |
> > > |Chinese idiom (4 character) |  2338    |  2379  |2362  |  2252 | 2252  | 2252   |  1876 | 1876  | 1876  |
> > > |Chinese idiom (multi-character)|  622    |  641  |629  |  576 | 576  | 576   |  334 | 334  | 334   |
> > > |English Word |  1261   |  1289  |1263  |  1076  | 1076   | 1076  |  842  | 842   | 842  |
> > > |English Idiom|  1237    |  1254  |  1244  |  1182 | 1182 | 1182  |  783 | 783 | 783  |
> > >
> > > [1] <https://arxiv.org/abs/2409.06851>
> > >
> > > [2] <https://arxiv.org/abs/2403.20330>
> > >
> > > > Concern 2: Ensuring fairness in evaluation.
> > >
> > > Thank you for your suggestion. In the evaluation, we strongly agree that a fair comparison is very important. Therefore, we try to select models with **the same number of parameters for comparison**, especially for open source models. We also try to ensure that we use **the same prompt for the same sub-task**. We apologize for the image drawing errors in our initial version of Figure 12, and have corrected the errors related to the prompt presentation in the new version. We promise that we maintained the **consistency of the prompt** across models during the review process, as we mentioned in the text. In addition, thank you for your suggestions regarding our newly added model's prompt, which we have added in Figure 12, as you can see in the latest version of the PDF (Page 26).

---

> > > ### Author Response · Authors · 2024-11-25
> > > **Response to Reviewer hb6G (Part3/3)**
> > >
> > > > Concern 3: It is unclear which of the existing evaluation methods best.
> > >
> > > Thank you for your suggestions. We apologize for your confusion about how the assessment metrics reflect MLLM's abilities, and hope that our next detailed explanation of the operation and specific moivation of each metric will make it clearer.
> > >
> > > - **Word-level (in Chinese idiom and English word) / Sentence-level (in English idiom)**: This is the most direct measure of MLLM's ability to fully understand the semantic information of the symbols in the image. When MLLM's output and the standard answer can be matched exactly at word-level or sentence-level (including structural matches), i.e., when MLLM successfully outputs the correct complete idiom, MLLM is considered to have answered the question correctly. At this level, we computed the associated precision, recall, and F-1 values. At this point, MLLM possesses both the understanding of individual emoji, and moreover the corresponding reasoning ability and text generation ability, which is the one that satisfies our initial motivation and **truly realizes the ability of unified visual-linguistic understanding**. Therefore, this is the most direct indicator of MLLM's ability.
> > >
> > > - **Character-level (in Chinese idiom and English word)/Word-level (in English idiom)**: due to the greater challenge of this benchmark, we found that without additional training, it is more difficult for MLLM to fully answer the correct and complete idiom. In order to better analyze which part of emoji-to-idiom comprehension is more challenging for MLLM, we evaluated at character-level/word-level and calculated Precision, recall, and F-1 values. Specifically, for English idiom, we computed BLEU-1 and BLEU-2 to better measure MLLM correctness at this level. Since we did not consider structural information in this segment, the Character/word-level metrics reflect more on MLLM's ability to understand individual emoji, due to which there are still a large number of emoji that just need to understand their meanings directly without additional reasoning. Therefore, MLLM's ability to **understand the emoji themselves is reflected** when MLLM receives a higher score in this item. If MLLM's score in the first item slips very significantly compared to the second item, we can conclude that MLLM possesses basic emoji comprehension skills but lacks further reasoning skills.
> > >
> > > - **Semantic similarity**: after we computed the character/word-level with exploring Chain-of-thought reasoning, we could not help but notice that sometimes MLLM is actually better at understanding individual emoji, predicting one or two characters correctly, but performs poorly at the full idiomorphic level. but poorer performance on the complete idiom level. There are even some MLLMs that correctly determine the meaning of each emoji during the CoT process, but when outputting the idiom, they output an idiom that has similar semantics but is completely different at the character level, resulting in serious semantic drift or even hallucination. Therefore, we added an extra step of calculating the metrics for the semantic similarity of the output response to the standard answer.
> > >   - We use LLM (specifically GPT-4o) as an expert to score the semantic similarity of the output response to the standard answer. The specific scoring criteria are as follows: scoring is done on a scale of 1-5, with 1 being completely dissimilar, 2 being relatively dissimilarity, 3 fairly similar, 4 being relatively similar, and 5 being completely similar. The specific prompt we use for scoring is: "Please measure the semantic similarity between the given standard answer and the model output on a scale of 1 to 5, where 1 means completely dissimilar, 2 means relatively dissimilarity, 3 means fairly similar, 4 means relatively similar, and 5 means completely similar. Output only a numerical score."
> > >   - The semantic similarity metrics **can be complemented with Character-level/word-level metrics**, both of which play an important role when the MLLM is unable to fully match the standard answer at the idiomorphic level. The semantic similarity metric focuses more on whether the answers output by the model are semantically similar to the standard answers, and does not focus on the understanding of individual emoji, but rather reflects an overall comprehension of the semantics of the text directly from the images.
> > >
> > > We provide a detailed description of the evaluation metrics and the different capabilities of MLLM they embody in the emoji2idiom, which helps researchers to use our benchmark and assess the specific capability bottlenecks of MLLM. We have updated the description of this section in the PDF, please see Page 21, Line 1085-1133, and Page 22, Line 1134-1144.
> > >
> > > Thank you again for your valuable comments, which have been very helpful in enhancing our work. We appreciate the effort you put into the review and hope that our response and revised PDF address your concerns.

---

> > > > ### Comment · Reviewer_hb6G · 2024-11-26
> > > >
> > > > I have carefully reviewed each of your responses and the parts related to paper revisions, especially the tables. They did address some of my doubts. However, I believe there are still significant issues in the benchmark evaluation section, so I will maintain my score. Thank you for your positive response.

---

> > > > > ### Author Response · Authors · 2024-11-26
> > > > >
> > > > > We sincerely thank you for the effort and time you have devoted to reviewing our work, as well as for your recognition of our contributions. We are delighted to hear that some of your concerns have been resolved through our revisions.
> > > > >
> > > > > That said, we understand you still have concerns about aspects of our evaluation phase. Given the extension of the discussion period, we are eager to further engage with you to address the issues you have noted regarding the benchmark evaluation. Your insights are invaluable, and we are committed to clarifying and improving wherever necessary. Once again, we deeply appreciate the generous support you have provided throughout this review process.

---

> > > > > ### Author Response · Authors · 2024-12-01
> > > > > **Friendly Follow-Up on Rebuttal and Response**
> > > > >
> > > > > As the rebuttal phase is coming to an end, I would like to thank you for your constructive feedback and positive discussions. We are pleased to hear that you recognize our work and that we have addressed most of your concerns, which is very encouraging for our research.
> > > > >
> > > > > In response to your remaining concerns about our evaluation, we have provided a detailed evaluation guide, highlighted in blue in the Appendix of the PDF. We hope this will help you better understand how to use our high-quality data for assessment.
> > > > >
> > > > > We sincerely hope to resolve any lingering doubts and gain your full endorsement. Your support is a great motivation for us to continue our research and further develop our dataset. We understand that your time and energy are very valuable, but with the deadline approaching, we would greatly appreciate your prompt feedback.

---

> > > > > ### Author Response · Authors · 2024-12-02
> > > > > **Friendly Follow-Up on Discussion about Evaluation**
> > > > >
> > > > > We would like to thank you for your feedback and positive discussions. We are pleased to hear that you recognize our work and that we **have addressed most of your concerns**, which is very encouraging for our research. In response to your remaining concerns about our evaluation, we have provided a **detailed evaluation guide, highlighted in blue in the Appendix of the PDF**. We hope this will help you better understand how to use our high-quality data for assessment.
> > > > >
> > > > > With the deadline approaching, we sincerely hope to resolve any lingering doubts and gain your full endorsement. Your support is a great motivation for us to continue our research and further develop our dataset!

---

> ### Author Response · Authors · 2024-11-25
> **Response to Reviewer hb6G (Part 1/3)**
>
> Thank you very much for your response. Your comments are very helpful in improving the quality of our work. We have addressed the concerns you raised with the following explanations and updated the relevant statements in the PDF. To make it easier for you to read, we have highlighted our latest changes using blue font and light blue highlighting.
>
> > Concern 1: Data leakage issues.
>
> First, we apologize for our inadequate literature research for data leakage. In the papers we listed, we focused primarily on the fact that their raw data was collected from the Internet, and did not further explore whether or not they had data leakage issues. After reading the two papers you provided, the work in them inspired us well and brought to our attention the relevance of our benchmark to these two pieces of work:
>
> - In LIME [1], we note that the authors' data filtering has three main steps, which coincides with the design of our emoji2idiom. We believe that emoji2idiom is **similar to LIME** in that both can do the job of avoiding tasks that are too hard or too easy, as well as **avoiding answer leakage**:
>   - The first step was to **categorize the difficulty level of the collected data and remove the very simple samples**. In emoji2idiom, on the other hand, our data and tasks are self-constructed brand new tasks, not filtered, and are naturally of a higher difficulty level (as evidenced by the results of the final evaluation, which showed that it was a challenging task). Therefore, as with LIME, there are **no very easy samples in our data**.
>   - The second step is to **remove Bad cases** that are too difficult.LIME uses a combination of GPT-4V and manual filtering to remove tasks that are too difficult among them. In emoji2idiom, we also removed idiomatic expressions that do not match human language usage and those with weak emoji-idiom correlations (please see Page 5 Line 219-223, Page 14 Line 740-755, Page 15 Line 756-764) in the human filtering to ensure that there are no samples in the data that are too difficult for MLLM to accomplish. In addition, we evaluated the ability of human experts on this data and found that human experts achieved an average score of 66.5 on Chinese idioms. This also proves that our benchmark is challenging while at the same time being a task that **can be completed** with the right level of difficulty.
>   - The third step is to **mitigate answer leakage**. LIME takes into account the answer leakage problem and pays more attention to tasks that require image comprehension ability, thus filtering out samples that can be answered by text only. In our task design, the answer of MLLM has high requirements on emoji image understanding and reasoning ability, and there is **no sample that can be answered directly without relying on images**. Therefore, we believe that we are similar to LIME in that we have both implemented the problem of **avoiding answer leakage** in our data construction.
>
> - Referring to the data construction method in MMStar [2], including its consideration of visual dependency and data leakage, we think that emoji2idiom data also exhibits better **visual dependency and minimal data leakage**.
>   - For visual dependency and data leakage, an **auto-filtering pipeline** is designed in MMStar that uses an open-source model for inference without image inputs and provides a 2-shot context to minimize rejection responses as a way to keep samples with hit counts less than or equal to 2 as being visually dependent and avoiding data leakage.
>   - Based on our previous analysis, we argue that our data satisfies the visual dependency, as well as the need to rely on images in order to reply. For further validation, we conduct a small batch of supplementary experiments with **reference to MMStar's filtering settings**. We randomly select 100 emoji-text pairs from each of the four tasks: four-character Chinese idioms, Chinese multi-character idioms, English words, and English idioms for testing. We experiment on the open-source model InternVL2-8B and the closed-source model GPT-4o with a 2-shot context example. Finally, it is found that **without image input, MLLM did not show a single correctly answered sample**, and thus scored $S_{wv} = 0$ without visual input. This proves the validity of our data.
>   - In addition, we note that in MMStar, since many of the original benchmark data had **multiple-choice question types**. Therefore, MLLM can sometimes answer correctly without visual input, and may just happen to guess the right option, rather than actually answering correctly without relying on vision. The use of all **open-ended questions** in our emoji2idiom data avoids this effect on the validity of the assessment caused by guessing the right option.

---

### Official Review · Reviewer_Bjya · 2024-11-02

**Soundness:** 3
**Presentation:** 3
**Contribution:** 2
**Rating:** 6
**Confidence:** 4

**Summary:**

The paper introduces Emoji2Idiom, a novel benchmark designed to evaluate the ability of Multimodal Large Language Models (MLLMs) to interpret emojis within images and map them to corresponding idioms in English and Chinese. The authors argue that this task necessitates complex reasoning skills across visual and linguistic modalities and highlight the challenges of harmonic mappings, fine-grained emoji understanding, and multi-emoji to one-word mapping. They propose a high-quality dataset comprising emoji-to-idiom mappings and present experimental results with several advanced MLLMs, providing insights into the models' current limitations and potential areas for improvement.

**Strengths:**

1. The paper introduces a creative and challenging task that requires MLLMs to interpret emojis within images and map them to complex idiomatic expressions. This task highlights a unique aspect of visual-language understanding that goes beyond traditional multimodal benchmarks, pushing models to engage in symbolic and cultural reasoning, which is closely aligned with human cognitive processing.

2. By including both English and Chinese idioms, the benchmark accommodates linguistic diversity and cultural depth, which is often underrepresented in multimodal tasks.  This cross-linguistic approach makes the dataset a valuable resource for exploring the generalization capabilities of MLLMs across different languages and idiomatic expressions, thus broadening its applicability and relevance for global AI applications.

**Weaknesses:**

1. Limited applicability to real-world scenarios: In practical applications, emojis are often input as text rather than images. The restriction of setting emojis as image input limits the benchmark's applicability to real-world contexts. Testing the performance with emojis as text input would more closely align with actual user behavior, thereby enhancing the benchmark's practical value.

2. Reliance on Harmonic Mapping for Semantics: The task design relies heavily on harmonic mappings, where emojis represent words or idioms based on similar sounds rather than direct meanings (e.g., phonetic similarities in Chinese). This reliance could constrain the model's understanding and may lead to brittle performance if deployed in real-world scenarios where direct meanings and context are more important than phonetic resemblance​.

**Questions:**

Is the visual input in the form of a single combined image of emojis, or is each emoji treated as an individual image?


Edit: I have read the author response and revised my score.

---

> ### Author Response · Authors · 2024-11-21
> **Response to reviewer Bjya (Part 1/2)**
>
> Thank you very much for your recognition of our work and your insightful comments. We hope to address your concerns through our response:
>
> > Weakness 1: Limited applicability to real-world scenarios.
>
> **Response**
>
> We apologize for causing you confusion about the reason for emoji as image input. We explain the human brain's mechanism for understanding emoji as visual symbols, the motivation for choosing emoji images, and MLLM's understanding of the real-world application of emoji, and hopefully address your concerns about the real-world applicability of emoji2idiom.
>
> - **For image input**: we agree that when people use emoji as input, it's usually in the form of text alongside other text. However, the emoji itself is a **visual symbol**. When understanding emoji, from the perspective of **human brain cognition**, people in real scenarios do not understand emoji directly as “text characters”, but rather **understand the visual information** of emoji from a visual perspective, so as to obtain the specific linguistic meaning represented by emoji. It makes sense to use emoji as image input to test the comprehension ability of LLM.
>
> - **For emoji as a representation of cryptographic symbols**: the main motivation of this benchmark is **not only** to focus on the use of LLMs that can understand emoji, but also to choose **an abstract symbol** such as emoji, which combines **both visual representations and text-specific semantic information**, to achieve a unified understanding of vision and language. That is, as we mentioned in the paper, to directly understand the semantic information of cryptographic symbols in images. This benchmark is different from previous benchmarks based on **VQA** forms, which represent visual and textual information **separately**, but directly condenses the features of **both into** the abstract symbols understanding of emoji. Therefore, we chose **the image form of emoji input**.
>
> - **For real-life application**: in real-life scenarios, people sometimes utilize emoji instead of textual characters for conversations, and although the input is in textual format at this point, what **LLM** really needs to have is **visual understanding**. With our proposed data, LLMs can be better challenged and trained to understand abstract symbols, which is more helpful to understand complex emoji representations of real-life scenarios.
>
> We have provided more details about the image input and benchmark motivation to eliminate potential confusion for readers, please note that you can see the lateset PDF version we uploaded especially the Page 1, Line 19-20; Page 2, Line 93-95; Page 3, Line123-128 in blue font.
>
> > Weakness 2: Reliance on Harmonic Mapping for Semantics.
>
> **Response**
>
> - **Harmonic Mapping Bias Mitigation**: First of all, for the **harmonic word mapping** dependency problem in the data, as we have described in Sec.3.2 (Page 5, Line 228-230) data annotation and filtering of the paper, we have taken into account that there may be a homophone mapping dependency problem when the model is further trained using emoji2idoim, i.e., an emoji is always mapped to a harmonic word, and this kind of data bias can cause model hallucination during the training. In order to **mitigate this data bias**, we have performed the homophone mapping statistics during filtering, and when an emoji maps to a homophone more than 10 times, we will replace it with another emoji or directly delete the corresponding data.
>
> - **Extension of cryptographic symbols and real-life scenario**: In addition, as we mentioned in Response to weakness 1, our benchmark is not concerned with the understanding of the meaning of **emoji itself**. Instead, we want to choose emoji, an abstract symbol with both **visual representation and specific textual semantics**, as the representative of this series of abstract cryptograms to **assess MLLM's ability** to make unified visual-language comprehension. With this motivation, we hope that the model can learn the ability of multimodal unified representation similar to that of the human brain, which can directly reason the semantic meaning of the corresponding text when seeing an image. The model's ability to understand and reason about specific semantic information of visually represented abstract symbols can be **transferred to other abstract symbols (e.g., chemical structure formulae, musical notation)**, thus providing a generalized ability to understand abstract symbols.

---

> > ### Comment · Reviewer_Bjya · 2024-11-26
> >
> > Thank you for your detailed reply.
> >
> > It makes sense to use emoji as image input to test the understanding of LLM.
> >
> > I will revise my comments and scores.

---

> > > ### Author Response · Authors · 2024-12-01
> > > **Friendly reminder to reviewer**
> > >
> > > We would like to express our sincere gratitude for your valuable feedback, effort, and time on our manuscript. We greatly appreciate your recognition of our work, which has been a significant encouragement for us.
> > >
> > > While you expressed a willingness to revise your comment and score after reviewing our rebuttal, the update hasn't yet been reflected. We understand that your time and energy are very valuable, and we do not wish to disturb you. However, with the rebuttal deadline approaching, we'd like to check if there were any remaining concerns or if there's anything more we could provide to assist in addressing your concerns and revising scores.
> > >
> > > Your unanimous recognition of the value and contribution of our research is greatly appreciated, and your recognition has deeply encouraged us for further research. Please feel free to let us know if there's anything specific you need from us. Thank you again for your dedication and comments on improving our manuscript.

---

> > > ### Author Response · Authors · 2024-12-02
> > > **Friendly reminder to reviewer Bjya**
> > >
> > > We would like to express our sincere gratitude for your effort on our manuscript. While you expressed a willingness to revise your comment and score, the score still remains unchanged. With the rebuttal deadline approaching, we'd like to know if there were any remaining concerns or if you might have inadvertently overlooked revising the score.
> > >
> > > We greatly appreciate your recognition of our work, which has been a significant encouragement for us. Please feel free to let us know if there's anything specific you need from us. Thank you again for your dedication and comments on improving our manuscript.

---

> ### Author Response · Authors · 2024-11-21
> **Response to reviewer Bjya (Part 2/2)**
>
> > Question 1:
> Is the visual input in the form of a single combined image of emojis, or is each emoji treated as an individual image?
>
> **Response:**
> In our input image, it is in a picture that has a series of emoji that form a sequence, and this emoji sequence will correspond to an idiom with a specific semantic meaning.
>
> > Question 2:
> Does the paper overlook evaluating the model’s performance when emojis are provided as text input?
>
> **Response:**
> Thanks for your suggestion. As we state in response to weakness 1, we do not just want LLM to understand the meaning of emoji, but we want LLM to realize that it understands a series of abstract cryptographic symbols represented by emoji that naturally have visual representations and specific textual semantics. Therefore we chose emoji in image form as the basis of the benchmark data, and we also believe that a review of emoji with textual input is not necessary in our work.

---

> ### Author Response · Authors · 2024-11-24
> **Looking Forward to Your Feedback and Discussion**
>
> Dear Reviewer Bjya,
>
> We sincerely appreciate the time and effort you have devoted to reviewing our paper and providing valuable feedback. Your insights have been instrumental in refining our work.  In our response, we have carefully addressed the concerns you raised, regarding the use of emoji images as input and the real-world applicability of the emoji2idiom benchmark. To provide clarity, we have included detailed explanations and submitted an updated PDF version of our manuscript.
>
> As the ICLR discussion phase is nearing its conclusion, we regret to note that we have not yet received your response to our rebuttal. We are eager to engage in further communication with you, as your expertise and perspective would significantly contribute to improving the quality of our paper.  We would be most grateful if you could share your thoughts at your earliest convenience. Thank you once again for your valuable contributions to this review process.
>
> Best regards,
>
> The Authors of Manuscript 6776

---

> ### Author Response · Authors · 2024-11-26
> **Friendly Follow-Up on Review Comments**
>
> Thank you for your valuable feedback and the effort you’ve dedicated to reviewing our work. We appreciate your recognition of our work and your insightful suggestions. While your comments have been edited, we notice that the scores remain unchanged. We kindly wonder if there are further concerns you’d like to discuss. Please let us know if any additional adjustments are needed. Once again, thank you for your time and support.

---

### Author Response · Authors · 2024-11-28
**Gentle Reminder to Response Due to PDF Update Deadline**

Dear Reviewers,

Thank you for your invaluable comments on our submission to ICLR. We would like to express our sincere gratitude for your valuable feedback and guidance on our submission.

As the deadline for updating the PDF is approaching under the latest ICLR discussion policy, we wonder if you have any further concerns regarding our response. This would help us address them by uploading a new PDF version before the deadline. We deeply appreciate your time and support. Please feel free to let us know if there’s anything further we can improve.

Warm regards,

The Authors of Manuscript 6776

---

### Note · Authors · 2024-12-16

**Comment:**

​Dear AC Chairs and Reviewers,

Sorry for interrupting you with this letter. After careful consideration by all of our authors, we would like to withdraw our manuscript titled 🤔Emoji2Idiom: Benchmarking Cryptic Symbol Understanding of Multimodal Large Language Models. The manuscript ID is 6776. Thank you for the effort and constructive comments, which greatly inspire us to further improve this paper.

Sincerely,
Authors of Paper 6776

**Withdrawal Confirmation:**

I have read and agree with the venue's withdrawal policy on behalf of myself and my co-authors.